# Sample-Efficient Learning of Stackelberg Equilibria in General-Sum Games

**Yu Bai**
Salesforce Research
yu.bai@salesforce.com

**Chi Jin**
Princeton University
chij@princeton.edu

**Huan Wang**
Salesforce Research
huan.wang@salesforce.com

**Caiming Xiong**
Salesforce Research
cxiong@salesforce.com

## Abstract

Real world applications such as economics and policy making often involve solving multi-agent games with two unique features: (1) The agents are inherently *asymmetric* and partitioned into leaders and followers; (2) The agents have different reward functions, thus the game is *general-sum*. The majority of existing results in this field focuses on either symmetric solution concepts (e.g. Nash equilibrium) or zero-sum games. It remains open how to learn the *Stackelberg equilibrium*—an asymmetric analog of the Nash equilibrium—in general-sum games efficiently from noisy samples. This paper initiates the theoretical study of sample-efficient learning of the Stackelberg equilibrium, in the bandit feedback setting where we only observe noisy samples of the reward. We consider three representative two-player general-sum games: bandit games, bandit-reinforcement learning (bandit-RL) games, and linear bandit games. In all these games, we identify a fundamental gap between the exact value of the Stackelberg equilibrium and its estimated version using finitely many noisy samples, which can not be closed information-theoretically regardless of the algorithm. We then establish sharp positive results on sample-efficient learning of Stackelberg equilibrium with value optimal up to the gap identified above, with matching lower bounds in the dependency on the gap, error tolerance, and the size of the action spaces. Overall, our results unveil unique challenges in learning Stackelberg equilibria under noisy bandit feedback, which we hope could shed light on future research on this topic.

## 1  Introduction

Real-world problems such as economic design and policy making can often be modeled as multi-agent games that involves two levels of thinking: The policy maker—as a player in this game—needs to reason about the other player's optimal behaviors given her decision, in order to inform her own optimal decision making. Consider for example the optimal taxation problem in the AI Economist [53], a game modeling a real-world social-economic system involving a *leader* (e.g. the government) and a group of interacting *followers* (e.g. citizens). The leader sets a tax rate which determines an economics-like game for the followers; the followers then play in this game with the objective to maximize their own reward (such as individual productivity). However, the goal of the leader is to maximize her own reward (such as overall equality) which is in general different from the followers' rewards, making these games *general-sum* [38]. Such two-level thinking appears broadly in other applications as well such as in automated mechanism design [11, 12], optimal auctions [10, 14], security games [43], reward shaping [25], and so on.

35th Conference on Neural Information Processing Systems (NeurIPS 2021).

Another key feature in such games is that the players are *asymmetric*, and they act in turns: the leader first plays, then the follower sees the leader's action and then adapts to it. This makes symmetric solution concepts such as Nash equilibrium [30] not always appropriate. A more natural solution concept for these games is the *Stackelberg equilibrium*: the leader's optimal strategy, assuming the followers play their best response to the leader [42, 13]. The Stackelberg equilbrium is often the desired solution concept in many of the aforementioned applications. Furthermore, it is of compelling interest to understand the learning of Stackelberg equilibria from *samples*, as it is often the case that we can only learn about the game through interactively deploying policies and observing the (noisy) feedback from the game [53]. This may be the case even when the game rules are perfectly known but not yet represented in a desired form, as argued in the line of work on empirical game theory [50, 48, 20].

Despite the motivations, theoretical studies of learning Stackelberg equilibria in general-sum games remain open, in particular when we can only learn from noisy samples of the rewards. A line of work provides guarantees for finding Stackelberg equilibria in general-sum games, but restricts attention to either the full observation setting (so that the exact game is observable) or with an exact best-response oracle [13, 27, 47, 34]. These results lay out a foundation for analyzing the Stackelberg equilibrium, but do not generalize to the bandit feedback setting in which the game can only be learned from random samples of the rewards. Another line of work considers the sample complexity of learning the Nash equilibrium in Markov games [35, 4, 5, 28, 51, 52], which also do not imply algorithms for finding the Stackelberg equilibrium in these games as the Nash is in general different from the Stackelberg equilibrium in general-sum games.

In this work, we study the sample complexity of learning Stackelberg equilibrium in general-sum games. We focus on general-sum games with two players (one leader and one follower), in which we wish to learn an approximate Stackelberg equilbrium for the leader from random samples. Our contributions can be summarized as follows.

- As a warm-up, we consider *bandit games* in which the two players play an action in turns and observe their own rewards. We identify a fundamental gap between the exact Stackelberg value and its estimated version from finite samples, which cannot be closed information-theoretically regardless of the algorithm (Section 3.1). We then propose a rigorous definition $\mathsf{gap}_\varepsilon$ for this gap, and show that it is possible to sample-efficiently learn the $(\mathsf{gap}_\varepsilon + \varepsilon)$-approximate Stackelberg equilibrium with $\widetilde{O}(AB/\varepsilon^2)$ samples, where $A, B$ are the number of actions for the two players. We further show a matching lower bound $\Omega(AB/\varepsilon^2)$ (Section 3.2). We also establish similar results for learning Stackelberg in simultaneous matrix games (Appendix B).

- We consider *bandit-RL games* in which the leader's action determines an episodic Markov Decision Process (MDP) for the follower. We show that a $(\mathsf{gap}_\varepsilon + \varepsilon)$ approximate Stackelberg equilibrium for the leader can be found in $\widetilde{O}(H^5 S^2 AB/\varepsilon^2)$ episodes of play, where $H, S$ are the horizon length and number of states for the follower's MDP, and $A, B$ are the number of actions for the two players (Section 4). Our algorithm utilizes recently developed reward-free reinforcement learning techniques to enable fast exploration for the follower within the MDPs.

- Finally, we consider *linear bandit games* in which the action spaces for the two players can be arbitrarily large, but the reward is a linear function of a $d$-dimensional feature representation of the actions. We design an algorithm that achieves $\widetilde{O}(d^2/\varepsilon^2)$ sample complexity upper bound for linear bandit games (Section 5). This only depends polynomially on the feature dimension instead of the size of the action spaces, and has at most an $\widetilde{O}(d)$ gap from the lower bound.

## 1.1 Related work

Since the seminal paper of [46], notions of equilibria in games and their algorithmic computation have received wide attention [see, e.g., 9, 41]. For the scope of this paper, this section focuses on reviewing results that related to learning Stackelberg equilibria.

**Learning Stackelberg equilibria in zero-sum games** The first category of results study two-player zero-sum games, where the rewards of the two players sum to zero. Most results along this line focus on the bilinear or convex-concave setting [see, e.g., 21, 32, 31, 37, 16], where the Stackelberg equilibrium coincide with the Nash equilibrium due to Von Neumann's minimax theorem [46]. Results for learning Stackelberg equilibria beyond convex-concave setting are much more recent,

with Rafique et al. [36], Nouiehed et al. [33] considering the nonconvex-concave setting, and Fiez et al. [15], Jin et al. [19], Marchesi et al. [29] considering the nonconvex-nonconcave setting. Marchesi et al. [29] provide sample complexity results for learning Stackelberg with infinite strategy spaces, using discretization techniques that may scale exponentially in the dimension without further assumptions on the problem structure.

We remark that a crucial property of zero-sum games is that any two strategies giving similar rewards for the follower will also give similar rewards for the leader. This is no longer true in general-sum games, and prevents most statistical results for learning Stackelberg equilibria in the zero-sum setting from generalizing to the general-sum setting.

**Learning Stackelberg equilibria in general-sum games**    The computational complexity of finding Stackelberg equilibria in games with simultaneous play ("computing optimal strategy to commit to") is studied in [13, 26, 47, 22, 1, 7]. These results assume full observation of the payoff function, and show that several versions of matrix games and extensive-form (multi-step) games admit polynomial time algorithms. Vasal [45] designs computationally efficient algorithms for computing Stackelberg in certain "conditionally independent controlled" Markov games.

Another line of work considers learning Stackelberg with a "best response oracle" [27, 6, 34], that returns the follower's exact best response strategy when a leader's strategy is queried. This oracle and the noisy reward oracle we assume are in general incomparable (cannot simulate each other regardless of the number of queries), and thus our sample complexity results do not imply each other. The recent work of Sessa et al. [39] proposes the StackelUCB algorithm to sample-efficiently learn a Stackelberg game where the opponent's response function has a linear structure in a certain kernel space, and the observation noise is added in the action space instead of the reward (thus a different noisy feedback model from ours).

Lastly, Fiez et al. [15] study the local convergence of first-order algorithms for finding Stackelberg equilibria in general-sum games. Their result also assumes exact feedback and do not allow sampling errors. The AI Economist [53] studies the optimal taxation problem by learning the Stackelberg equilibrium via a two-level reinforcement learning approach.

**Learning equilibria in Markov games**    A recent line of results [4, 5, 51, 52] consider learning Markov games [40]—a generalization of Markov decision process to the multi-agent setting. We remark that all three settings studied in this paper can be cast as special cases of general-sum Markov games, which is studied by [28]. In particular, Liu et al. [28] provides sample complexity guarantees for finding Nash equilibria, correlated equilibria, or coarse correlated equilibria of the general-sum Markov games. These Nash-finding algorithms are related to our setting, but do not imply results for learning Stackelberg (see Section 3.2 and Appendix C.5 for detailed discussions).

## 2    Preliminaries

**Bandit games**    A general-sum two-player bandit game can be described by a tuple $M = (\mathcal{A}, \mathcal{B}, r_1, r_2)$, which defines the following game played by two players, a *leader* and a *follower*:

- The leader plays an action $a \in \mathcal{A}$, with $|\mathcal{A}| = A$.
- The follower sees the action played by the leader, and plays an action $b \in \mathcal{B}$, with $|\mathcal{B}| = B$.
- The follower observes a (potentially random) reward $r_2(a, b) \in [0, 1]$. The leader also observes her own reward $r_1(a, b) \in [0, 1]$.

Note that this is a special case of a general-sum turn-based Markov game with two steps [40, 4]. This game is also a turn-based variant of the simultaneous matrix game considered in [13, 27] (for which we also provide results in Appendix B).

**Best response, Stackelberg equilibrium**    Let $\mu_i(a, b) := \mathbb{E}[r_i(a, b)]$ $(i = 1, 2)$ denote the mean rewards. For each leader action $a$, the *best response set* $\mathsf{BR}_0(a)$ is the set of follower actions that maximize $\mu_2(a, \cdot)$:

$$\mathsf{BR}_0(a) := \left\{ b : \mu_2(a, b) = \max_{b' \in \mathcal{B}} \mu_2(a, b') \right\}. \tag{1}$$

Given the best-response set $\mathsf{BR}_0(a)$, we define the function $\phi_0 : \mathcal{A} \to [0, 1]$ as the leader's value function when the follower plays the worst-case best response (henceforth the "exact $\phi$-function"):

$$\phi_0(a) := \min_{b \in \mathsf{BR}_0(a)} \mu_1(a, b), \tag{2}$$

This is the value function for the leader action $a$, assuming the follower plays the best response to $a$ and breaks ties in the best response set against the leader's favor. This is known as *pessimistic tie breaking* and provides a worst-case guarantee for the leader [13]. We remark that here restricting $b$ to pure strategies (deterministic actions) is without loss of generality, as there is at least one pure strategy that maximizes $\mu_2(a, \cdot)$ and (among the maximizers) minimize $\mu_1(a, \cdot)$, among all mixed strategies.

The Stackelberg Equilibrium (henceforth also "Stackelberg") for the leader is the "best response to the best response", i.e. any action $a_\star$ that maximizes $\phi_0$ [42]:

$$a_\star \in \arg\max_{a \in \mathcal{A}} \phi_0(a). \tag{3}$$

We are interested in finding approximate solutions to the Stackelberg equilibrium, that is, an action $\widehat{a}$ that approximately maximizes $\phi_0(a)$. Note that as the leader's action is seen by the follower, it suffices to only consider pure strategies for the leader too (this is equivalent to the "optimal committment to pure strategies" problem of [13]). We also remark that, while we consider pessimistic tie breaking (definition (2)) in this paper, similar results hold in the optimistic setting as well in which the follower breaks ties in favor of the leader. We defer the statements and proofs of these results to Appendix A.

**Real-world example** (AI Economist): Consider the following (simplified) optimal taxation problem in the AI Economist [53] as an example of a bandit game. The government (leader) determines the tax rate $a \in \mathcal{A}$ for the follower. The citizen (follower) then chooses the amount of labor $b \in \mathcal{B}$ she wishes to perform. The rewards for the two players are different in general: For example, the citizen's reward $r_2(a, b)$ can be her post-tax income per labor, and the government's reward $r_1(a, b)$ can be a weighted average of its tax income and some measure of the citizen's welfare (e.g. not too much labor). We remark that the actual AI Economist is more similar to a *bandit-RL game* where the follower plays sequentially in an MDP determined by the leader, which we study in Section 4.

**Sample-efficient learning with bandit feedback**    In this paper we consider the *bandit feedback* setting, that is, the algorithm cannot directly observe the mean rewards $\mu_1(\cdot, \cdot)$ and $\mu_2(\cdot, \cdot)$, and can only query $(a, b)$ and obtain random samples $(r_1(a, b), r_2(a, b))$. Our goal is to determine the number of samples in order to find an approximate maximizer of $\phi_0(a)$.

Note that the bandit feedback setting assumes observation noise in the rewards. As we will see in Section 3, this noise turns out to bring in a fundamental challenge for learning Stackelberg equilibria that is not present in (and thus not directly solved by) existing work on learning Stackelberg, which assumes either exact observation of the mean rewards [13, 26], or the best-response oracle that can query $a$ and obtain the exact best response set $\mathsf{BR}_0(a)$ [27, 34].

**Markov decision processes**    We also present the basics of a Markov Decision Processes (MDPs), which will be useful for the bandit-RL games in Section 4. We consider episodic MDPs defined by a tuple $(H, \mathcal{S}, \mathcal{B}, d^1, \mathbb{P}, r)$, where $H$ is the horizon length, $\mathcal{S}$ is the state space, $\mathcal{B}$ is the action space[1], $\mathbb{P} = \{\mathbb{P}_h(\cdot|s, b) : h \in [H], s \in \mathcal{S}, b \in \mathcal{B}\}$ is the transition probabilities, and $r = \{r_h : \mathcal{S} \times \mathcal{B} \to [0, 1], h \in [H]\}$ are the (potentially random) reward functions. A policy $\pi = \{\pi_h^b(\cdot|s) \in \Delta_\mathcal{B} : h \in [H], s \in \mathcal{S}\}$ for the player is a set of probability distributions over actions given the state.

In this paper we consider the exploration setting as the protocol of interacting with MDPs, similar as in [3, 17]. The learning agent is able to play episodes repeatedly, where in each episode at step $h \in \{1, \dots, H\}$, the agent observes state $s_h \in \mathcal{S}$, takes an action $b_h \sim \pi_h(\cdot|s_h)$, observes her reward $r_h = r_h(s_h, b_h) \in [0, 1]$, and transits to the next state $s_{h+1} \sim \mathbb{P}_h(\cdot|s_h, b_h)$. The initial state is received from the MDP: $s_1 \sim d^1(\cdot)$. The overall value function (return) of a policy $\pi$ is defined as $V(\pi) := \mathbb{E}_\pi\left[\sum_{h=1}^H r_h(s_h, b_h)\right]$.

---

[1]The notation $\mathcal{B}$ indicates that the MDP is played by the follower (cf. Section 4); we reserve $\mathcal{A}$ as the leader's action space in this paper.

## 3 Warm-up: bandit games

### 3.1 Hardness of maximizing $\phi_0$ from samples

Given the exact $\phi$-function $\phi_0$ (2), a natural notion of approximate Stackelberg equilibrium is to find an action $\widehat{a}$ that is $\varepsilon$ near-optimal for maximizing $\phi_0$:

$$\phi_0(\widehat{a}) \geq \max_{a \in \mathcal{A}} \phi_0(a) - \varepsilon. \tag{4}$$

However, the following lower bound shows that, in the worst case, it is hard to find such $\widehat{a}$ from finite samples.

**Theorem 1** ($\Omega(1)$ lower bound for maximizing $\phi_0$). *For any sample size $n$ and any algorithm for maximizing $\phi_0$ that outputs an action $\widehat{a} \in \mathcal{A}$, there exists a bandit game with $A = B = 2$ on which the algorithm must suffer from $\Omega(1)$ error with probability at least $1/3$:*

$$\phi_0(\widehat{a}) \leq \max_{a \in \mathcal{A}} \phi_0(a) - 1/2.$$

Theorem 1 shows that, no matter how large the sample size $n$ is, any algorithm in the worst-case have to suffer from an $\Omega(1)$ lower bound for maximizing $\phi_0$ (i.e. determining the Stackelberg equilibrium for the leader). This result stems from a *hardness of determining the best response* $\mathsf{BR}_0(a)$ *exactly* from samples. (See Table 2 for the construction of the hard instance and Appendix C.1 for the full proof of Theorem 1.) This is in stark contrast to the standard $1/\sqrt{n}$ type learning result in finding other solution concepts such as the Nash equilibrium [4, 28], and suggests a new fundamental challenge to learning Stackelberg equilibrium from samples.

### 3.2 Learning Stackelberg with value optimal up to gap

The lower bound in Theorem 1 shows that approximately maximizing $\phi_0$ is information-theoretically hard. Motivated by this, we consider in turn a slightly relaxed notion of optimality, in which we consider maximizing $\phi_0$ only up to the *gap* between $\phi_0$ and its counterpart using $\varepsilon$-approximate best responses. More concretely, define the $\varepsilon$-approximate versions of the best response set and $\phi$-function as

$$\phi_\varepsilon(a) := \min_{b \in \mathsf{BR}_\varepsilon(a)} \mu_1(a, b),$$

$$\mathsf{BR}_\varepsilon(a) := \left\{ b \in \mathcal{B} : \mu_2(a, b) \geq \max_{b'} \mu_2(a, b') - \varepsilon \right\}.$$

These definitions are similar to the vanilla $\mathsf{BR}_0$ and $\phi_0$ in (1) and (2), except that we allow any $\varepsilon$-approximate best response to be considered as a valid response to the leader action. Observe we always have $\mathsf{BR}_\varepsilon(a) \supseteq \mathsf{BR}_0(a)$ and $\phi_\varepsilon(a) \leq \phi_0(a)$. We then define the *gap* of the game for any $\varepsilon \in (0, 1)$ as

$$\mathsf{gap}_\varepsilon := \max_{a \in \mathcal{A}} \phi_0(a) - \max_{a \in \mathcal{A}} \phi_\varepsilon(a) \geq 0. \tag{5}$$

This $\mathsf{gap}_\varepsilon$ is discontinuous in $\varepsilon$ in general, and can be as large as $\Theta(1)$ for any $\varepsilon > 0$ without additional assumptions on the relation between $\mu_1$ and $\mu_2$[2]. See Figure 1 for an illustration for a typical $\max_{a \in \mathcal{A}} \phi_\varepsilon(a)$ against $\varepsilon$.

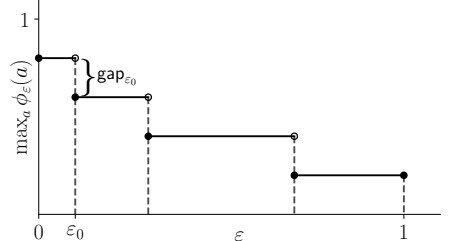

Figure 1: Illustration of $\max_{a \in \mathcal{A}} \phi_\varepsilon(a)$ and $\mathsf{gap}_\varepsilon$ as a function of $\varepsilon$. The quantity $\mathsf{gap}_{\varepsilon_0}$ can be $\Omega(1)$ for arbitrarily small $\varepsilon_0$.

With the definition of the gap, we are now ready to state our main result, which shows that it is possible to sample-efficiently learn Stackelberg Equilibria with value up to $(\mathsf{gap}_\varepsilon + \varepsilon)$. The proof can be found in Appendix C.3.

---

[2]This $\mathsf{gap}_\varepsilon$ could be small when $\mu_1$ and $\mu_2$ have certain relations, such as zero-sum or cooperative structure. See Appendix C.6 for more discussions.

---

**Algorithm 1** Learning Stackelberg in bandit games

---

**Require:** Target accuracy $\varepsilon > 0$.

**set** $N \leftarrow C \log(AB/\delta)/\varepsilon^2$ for some constant $C > 0$.

1: Query each $(a, b) \in \mathcal{A} \times \mathcal{B}$ for $N$ times and obtain $\{r_1^{(j)}(a, b), r_2^{(j)}(a, b)\}_{j=1}^N$.

2: Construct empirical estimates of the means $\widehat{\mu}_i(a, b) = \frac{1}{N} \sum_{j=1}^N r_i^{(j)}(a, b)$ for $i = 1, 2$.

3: Construct approximate best response sets and values for all $a \in \mathcal{A}$:

$$\widehat{\phi}_{3\varepsilon/4}(a) := \min_{b \in \widehat{\mathsf{BR}}_{3\varepsilon/4}(a)} \widehat{\mu}_1(a, b), \quad \text{where} \quad \widehat{\mathsf{BR}}_{3\varepsilon/4}(a) := \left\{ b : \widehat{\mu}_2(a, b) \geq \max_{b' \in \mathcal{B}} \widehat{\mu}_2(a, b') - 3\varepsilon/4 \right\}.$$

4: Output $(\widehat{a}, \widehat{b})$, where $\widehat{a} = \arg\max_{a \in \mathcal{A}} \widehat{\phi}_{3\varepsilon/4}(a)$, $\widehat{b} = \arg\min_{b \in \widehat{\mathsf{BR}}_{3\varepsilon/4}(\widehat{a})} \widehat{\mu}_1(\widehat{a}, b)$.

---

**Theorem 2** (Learning Stackelberg in bandit games). *For any bandit game and $\varepsilon \in (0, 1)$, Algorithm 1 outputs $(\widehat{a}, \widehat{b})$ such that with probability at least $1 - \delta$,*

$$\phi_0(\widehat{a}) \geq \phi_{\varepsilon/2}(\widehat{a}) \geq \max_{a \in \mathcal{A}} \phi_0(a) - \mathtt{gap}_\varepsilon - \varepsilon,$$

$$\mu_2(\widehat{a}, \widehat{b}) \geq \max_{b' \in \mathcal{B}} \mu_2(\widehat{a}, b') - \varepsilon$$

*with $n = \widetilde{O}(AB/\varepsilon^2)$ samples, where $\widetilde{O}(\cdot)$ hides log factors. Further, the algorithm runs in $O(n) = \widetilde{O}(AB/\varepsilon^2)$ time.*

**Implications; Overview of algorithm** Theorem 2 shows that it is possible to learn $\widehat{a}$ that maximizes $\phi_0(a)$ up to $(\mathtt{gap}_\varepsilon + \varepsilon)$ accuracy, using $\widetilde{O}(AB/\varepsilon^2)$ samples. The quantity $\mathtt{gap}_\varepsilon$ is not bounded and can be as large as $\Theta(1)$ for any $\varepsilon$ (see Lemma C.1 for a formal statement); however the gap is non-increasing as we decrease $\varepsilon$. In the situation where for every $a$ the best follower action for $\mu_2(a, \cdot)$ is at least $\varepsilon_0$-better than the second best action, then for $\varepsilon < \varepsilon_0$ we have $\mathtt{gap}_\varepsilon = 0$ and Theorem 2 implies an $\varepsilon$-optimal Stackelberg guarantee. In general, Theorem 2 presents a "best-effort" positive result for learning Stackelberg under this relaxed notion of optimality. To the best of our knowledge, this is the first result for sample-efficient learning of Stackelberg equilibrium in general-sum games with noisy bandit feedbacks. We remark that Theorem 2 also provides a near-optimality guarantee for $\phi_{\varepsilon/2}(\widehat{a})$ which is slightly stronger than $\phi_0$ (since $\phi_{\varepsilon/2}(\widehat{a}) \leq \phi_0(\widehat{a})$), and guarantees the learned $\widehat{b}$ is indeed an $\varepsilon$-approximate best response of $\widehat{a}$.

From a more practical point of view, Theorem 2 (and our later results on bandit-RL games and linear bandit games) spells out concretely the sample size required to learn an $\varepsilon$-approximate Stackelberg, in terms of the scaling with problem parameters. For instance, in the AI Economist example, $A$ is the number of tax rate choices for the government, and $B$ is the number of actions for the citizen, and our results show that there exist an algorithm with sample complexity polynomial in $A$, $B$, and $1/\varepsilon$.

The main step in Algorithm 1 is to construct approximate best response sets $\widehat{\mathsf{BR}}_{3\varepsilon/4}(a)$ for all $a \in \mathcal{A}$ based on the empirical estimates of the rewards. Through concentration, we argue that $\widehat{\mathsf{BR}}_{3\varepsilon/4}(a)$ is a good approximation of the true best response sets in the sense that $\mathsf{BR}_{\varepsilon/2}(a) \subseteq \widehat{\mathsf{BR}}_{3\varepsilon/4}(a) \subseteq \mathsf{BR}_\varepsilon(a)$ holds for all $a \in \mathcal{A}$, from which the Stackelberg guarantee follows.

**Irreducibility to Nash-finding algorithms** We also remark that our bandit game is equivalent to a turn-based general-sum Markov game with two steps, $A$ states, and $(A, B)$ actions [4]. Further, the Stackelberg equilibrium $a_\star$ along with a follower policy that plays the best response (with pessimistic tie-breaking) constitutes a Nash equilibrium for that Markov game (see Appendix C.5 for a formal statement and proof). However, existing Nash-finding algorithms for general-sum Markov games such as `Multi-Nash-VI` [28] *do not* imply an algorithm for finding the Stackelberg equilibrium. This is because general-sum games have multiple Nash equilibria (with different values) in general [38], and these existing Nash-finding algorithms cannot pre-specify which Nash to output.

**Lower bound** We accompany Theorem 2 by an $\Omega(AB/\varepsilon^2)$ sample complexity lower bound, showing that Theorem 2 achieves the optimal sample complexity up to logarithmic factors.

**Theorem 3** (Lower bound for bandit games). *There exists an absolute constant $c > 0$ such that the following holds. For any $\varepsilon \in (0, c)$, $g \in [0, c)$, any $A, B \geq 3$, and any algorithm that queries $N \leq c[AB/\varepsilon^2]$ samples and outputs an estimate $\widehat{a} \in \mathcal{A}$, there exists a bandit game $M$ on which $\mathsf{gap}_\varepsilon = g$ and the algorithm suffers from $(g + \varepsilon)$ error:*

$$\phi_{\varepsilon/2}(\widehat{a}) \leq \phi_0(\widehat{a}) \leq \max_{a \in \mathcal{A}} \phi_0(a) - g - \varepsilon$$

*with probability at least $1/3$.*

This lower bound shows the tightness of Theorem 2, and suggests that $(\mathsf{gap}_\varepsilon + \varepsilon)$ suboptimality is perhaps a sensible learning goal, as for any algorithm and any value of $g \geq 0$ there exists a game with $\mathsf{gap}_\varepsilon = g$, on which the algorithm has to suffer from $(g + \varepsilon)$ error, if the number of samples is at most $O(AB/\varepsilon^2)$. The proof of Theorem 3 is deferred to Appendix C.4.

## 4 Bandit-RL games

In this section, we investigate learning Stackelberg equilibrium in bandit-RL games, in which each leader's action determines an episodic Markov Decision Process (MDP) for the follower. This setting extends the two-player bandit games by allowing the follower to play sequentially, and has strong practical motivations in particular in policy making problems involving sequential plays for the follower, such as the optimal taxation problem in the AI Economist [53].

**Setting** A bandit-RL game is described by the leader's action set $\mathcal{A}$ (with $|\mathcal{A}| = A$), and a family of MDPs $M = \{M^a : a \in \mathcal{A}\}$. Each leader action $a \in \mathcal{A}$ determines an episodic MDP $M^a = (H, \mathcal{S}, \mathcal{B}, \mathbb{P}^a, r_{1,h}(a, \cdot, \cdot), r_{2,h}(a, \cdot, \cdot))$ that contains $H$ steps, $S$ states, $B$ actions, with two reward functions $r_1$ and $r_2$. In each episode of play,

- The leader plays action $a \in \mathcal{A}$.
- The follower sees this action and enters the MDP $M^a$. She observes the deterministic[3] initial state $s_1$, and plays in $M^a$ with exploration feedback for one episode.
- While the follower plays in the MDP, she observes reward $r_{2,h}(a, s_h, b_h)$, whereas the leader also observes her own reward $r_{1,h}(a, s_h, b_h)$.

We let $\pi^b$ denote a policy for the follower, and let $V_1(a, \pi^b)$ and $V_2(a, \pi^b)$ denote its value functions (in $M^a$) for the leader and the follower respectively.

Similar as in bandit games, we define the $\varepsilon$-approximate best-response set $\mathsf{BR}_\varepsilon(a)$ and the $\varepsilon$-approximate $\phi$-function $\phi_\varepsilon(a)$ for all $\varepsilon \geq 0$ as

$$\phi_\varepsilon(a) := \min_{\pi^b \in \mathsf{BR}_\varepsilon(a)} V_1(a, \pi^b), \quad \text{where} \quad \mathsf{BR}_\varepsilon(a) := \left\{ \pi^b : V_2(a, \pi^b) \geq \max_{\widetilde{\pi}^b} V_2(a, \widetilde{\pi}^b) - \varepsilon \right\}.$$

Define $\mathsf{gap}_\varepsilon = \max_{a \in \mathcal{A}} \phi_0(a) - \max_{a \in \mathcal{A}} \phi_\varepsilon(a)$ similarly as in (5). We are interested in the number of episodes in order to find a $(\mathsf{gap}_\varepsilon + \varepsilon)$ near-optimal Stackelberg equilibrium.

### 4.1 Algorithm description

At a high level, our algorithm for bandit-RL games is similar as for bandit games – query each leader action $a \in \mathcal{A}$ sufficiently many times, let the follower learn the best response (i.e. best policy for the MDP $M^a$) for each $a \in \mathcal{A}$, and then choose the leader action that maximizes the best response value function. This requires solving

$$\arg\max_{a \in \mathcal{A}} \phi_{3\varepsilon/4}(a) := \arg\max_{a \in \mathcal{A}} \min_{\pi^b \in \widehat{\mathsf{BR}}_{3\varepsilon/4}(a)} \widehat{V}_1(a, \pi^b),$$

$$\widehat{\mathsf{BR}}_{3\varepsilon/4}(a) := \left\{ \pi^b : \widehat{V}_2(a, \pi^b) \geq \max_{\widetilde{\pi}^b} \widehat{V}_2(a, \widetilde{\pi}^b) - 3\varepsilon/4 \right\},$$

(6)

---

[3]The general case where $s_1$ is stochastic reduces to the deterministic case by adding a step $h = 0$ with a single deterministic initial state $s_0$, which only increases the horizon of the game by 1.

---

**Algorithm 2** Learning Stackelberg in bandit-RL games

---

**Require:** Target accuracy $\varepsilon > 0$.

1: **for** $a \in \mathcal{A}$ **do**
2:     Let the leader pull arm $a \in \mathcal{A}$ and the follower run the `Reward-Free RL-Explore` algorithm for $N \leftarrow \widetilde{O}(H^5 S^2 B/\varepsilon^2 + H^7 S^4 B/\varepsilon)$ episodes, and obtain model estimate $\widehat{M}^a$.
3:     Let $(\widehat{V}_1(a, \cdot), \widehat{V}_2(a, \cdot))$ denote the value functions for the model $\widehat{M}^a$.
4:     Compute the empirical best response value $\widehat{V}_2^\star(a) := \max_{\pi^b} \widehat{V}_2(a, \pi^b)$ by any optimal planning algorithm (e.g. value iteration) on the empirical MDP $\widehat{M}^a$.
5:     Solve the following program

$$\text{minimize}_{\pi^b} \ \widehat{V}_1(a, \pi^b) \ \text{ s.t. } \ \pi^b \in \widehat{\mathsf{BR}}_{3\varepsilon/4}(a) := \left\{ \pi^b : \widehat{V}_2(a, \pi^b) \geq \widehat{V}_2^\star(a) - 3\varepsilon/4 \right\} \tag{7}$$

    by subroutine $(\widehat{\pi}^{b,(a)}, \widehat{\phi}_{3\varepsilon/4}(a)) \leftarrow \texttt{WorstCaseBestResponse}(\widehat{M}^a, \widehat{V}_2^\star(a) - 3\varepsilon/4)$.

**output** $(\widehat{a}, \widehat{\pi}^b)$ where $\widehat{a} \leftarrow \arg\max_{a \in \mathcal{A}} \widehat{\phi}_{3\varepsilon/4}(a)$ and $\widehat{\pi}^b \leftarrow \widehat{\pi}^{b,(\widehat{a})}$.

---

where $\widehat{V}_1$ and $\widehat{V}_2$ are empirical estimates of the true value functions.

Two technical challenges emerge as we instantiate (6). First, the follower not only needs to find her own best policy during the exploration phase, but also has to accurately estimate both her own and the leader's reward over the entire approximate best response set $\widehat{\mathsf{BR}}_{3\varepsilon/4}$ so as to make sure the estimates $\widehat{V}_i(a, \pi^b)$ $(i = 1, 2)$ reliable. Standard fast PAC-exploration algorithms such as those in [3, 17] do not provide such guarantees. We resolve this by applying the *reward-free learning algorithm* of Jin et al. [18] for the follower to explore the environment efficiently while providing value concentration guarantees for multiple rewards and policies. We remark that we slightly generalize the guarantees of [18] to the situation where the rewards are random and have to be estimated from samples.

Second, the problem $\min_{\pi^b \in \widehat{\mathsf{BR}}_{3\varepsilon/4}(a)} \widehat{V}_1(a, \pi^b)$ in (6) requires minimizing a value function over the near-optimal policy set of another value function. We build on the linear programming reformulation in the constrained MDP literature [2] to translate (6) into a linear program `WorstCaseBestResponse`, which adopts efficient solution in $\text{poly}(HSB)$ time [8]. (the description of this subroutine can be found in Algorithm 8 in Appendix D.1). Our full algorithm is described in Algorithm 2.

### 4.2 Main result

We now state our theoretical guarantee for Algorithm 2. The proof can be found in Appendix D.2.

**Theorem 4** (Learning Stackelberg in bandit-RL games)**.** *For any bandit-RL game and sufficiently small $\varepsilon \leq O(1/H^2 S^2)$, Algorithm 2 with $n = \widetilde{O}(H^5 S^2 AB/\varepsilon^2 + H^7 S^4 AB/\varepsilon)$ episodes of play can return $(\widehat{a}, \widehat{\pi}^b)$ such that with probability at least $1 - \delta$,*

$$\phi_0(\widehat{a}) \geq \phi_{\varepsilon/2}(\widehat{a}) \geq \max_{a \in \mathcal{A}} \phi_0(a) - \mathtt{gap}_\varepsilon - \varepsilon,$$

$$V_2(\widehat{a}, \widehat{\pi}^b) \geq \max_{\widetilde{\pi}^b} V_2(\widehat{a}, \widetilde{\pi}^b) - \varepsilon,$$

*where $\widetilde{O}(\cdot)$ hides $\log(HSAB/\delta\varepsilon)$ factors. Further, the algorithm runs in $\text{poly}(HSAB/\delta\varepsilon)$ time.*

**Sample complexity, relationship with reward-free RL** Theorem 4 shows that for bandit-RL games, the approximate Stackelberg Equilibrium (with value optimal up to $\mathtt{gap}_\varepsilon + \varepsilon$) can be efficiently found with polynomial sample complexity and runtime. In particular, (for small $\varepsilon$) the leading term in the sample complexity scales as $\widetilde{O}(H^5 S^2 AB/\varepsilon^2)$. Since bandit-RL games include bandit games as a special case, the $\Omega(AB/\varepsilon^2)$ lower bound for bandit games (Theorem 3) apply here and implies that the $A, B$ dependence in Theorem 4 is tight, while the $H$ dependence may be slightly suboptimal.

We also remark the learning goal for the follower in our bandit-RL game is a new RL setting in between the single-reward setting and the full reward-free setting, for which the optimal $S$ dependence is currently unclear. In the single-reward setting, existing fast exploration algorithms such as UCBVI [3] only require linear in $S$ episodes for finding a near-optimal policy. In contrast, in the full reward-free

---

**Algorithm 3** Learning Stackelberg in linear bandit games

---

**Require:** Target accuracy $\varepsilon > 0$.

1: Find $(\mathcal{K}, \rho) \leftarrow \mathtt{CoreSet}(\Phi)$ (cf. (10)). Let $\mathcal{K} = \{(a_j, b_j) : 1 \leq j \leq K\}$ where $K = |\mathcal{K}|$.

2: Query each $(a_j, b_j)$ for $N = O(d \log(d/\delta)/\varepsilon^2)$ times. Let $(\widehat{\mu}_{1,j}, \widehat{\mu}_{2,j})$ denote the empirical mean of the observed rewards over the $N$ queries.

3: Estimate $(\theta_1^\star, \theta_2^\star)$ via weighted least squares

$$\widehat{\theta}_i := \arg \min_{\theta \in \mathbb{R}^d} \sum_{i=1}^{K} \rho(a_j, b_j) \big(\phi(a_j, b_j)^\top \theta_i - \widehat{\mu}_{i,j}\big)^2, \quad i = 1, 2. \tag{9}$$

4: Construct approximate best response sets and values for all $a \in \mathcal{A}$:

$$\widehat{\mathsf{BR}}_{3\varepsilon/4}(a) := \left\{ b : \phi(a, b)^\top \widehat{\theta}_2 \geq \max_{b' \in \mathcal{B}} \phi(a, b')^\top \widehat{\theta}_2 - 3\varepsilon/4 \right\},$$

$$\widehat{\phi}_{3\varepsilon/4}(a) := \min_{b \in \widehat{\mathsf{BR}}_{3\varepsilon/4}(a)} \phi(a, b)^\top \widehat{\theta}_1.$$

5: Output $(\widehat{a}, \widehat{b})$, where $\widehat{a} = \arg \max_{a \in \mathcal{A}} \widehat{\phi}_{3\varepsilon/4}(a), \widehat{b} = \arg \min_{b \in \widehat{\mathsf{BR}}_{3\varepsilon/4}(\widehat{a})} \phi(\widehat{a}, b)^\top \widehat{\theta}_1.$

---

setting (follower wants accurate estimation of any reward), it is known $\Omega(S^2)$ is unavoidable [18]. Our setting poses a unique challenge in between: The follower wishes to accurately estimate both $r_1, r_2$ on all near-optimal policies for $r_2$. This further renders recent linear in $S$ algorithms for reward-free learning with finitely many rewards [28] not applicable here, as they only guarantee accurate estimation of each reward on near-optimal policies for *that* reward. We believe the optimal sample complexity for bandit-RL games is an interesting open question.

## 5 Linear bandit games

**Setting** We consider a bandit game with action space $\mathcal{A}, \mathcal{B}$ that are finite but potentially arbitrarily large, and assume in addition that the reward functions has a linear form

$$r_1(a, b) = \phi(a, b)^\top \theta_1^\star + z_1, \quad r_2(a, b) = \phi(a, b)^\top \theta_2^\star + z_2, \tag{8}$$

where $\phi : \mathcal{A} \times \mathcal{B} \to \mathbb{R}^d$ is a $d$-dimensional feature map, $\theta_1^\star, \theta_2^\star \in \mathbb{R}^d$ are unknown ground truth parameters for the rewards, and $z_1, z_2$ are random noise which we assume are mean-zero and 1-sub-Gaussian. Let $\Phi := \{\phi(a, b) : (a, b) \in \mathcal{A} \times \mathcal{B}\}$ denote the set of all possible features. For linear bandit games, we define $\mathtt{gap}_\varepsilon$ same as definition (5) for bandit games.

**Algorithm and guarantee** We present our algorithm for linear bandit games in Algorithm 3. Compared with our Algorithm 1 for bandit games, Algorithm 3 takes advantage of the linear structure through the following important modifications: (1) Rather than querying every action pair, we only query $(a, b)$ in a *core set* $\mathcal{K}$ found through the following subroutine

$\mathtt{CoreSet}(\Phi) := (\mathcal{K}, \rho)$ where $\mathcal{K} \subset \mathcal{A} \times \mathcal{B}$, $\rho \in \Delta_{\mathcal{K}}$, such that

$$\max_{\phi \in \Phi} \phi^\top \Big( \sum_{(a,b) \in \mathcal{K}} \rho(a, b) \phi(a, b) \phi(a, b)^\top \Big)^{-1} \phi \leq 2d \quad \text{and} \quad K = |\mathcal{K}| \leq 4d \log \log d + 16. \tag{10}$$

Such a core set is guaranteed to exist for any finite $\Phi$ [24, Theorem 4.4], and can be found efficiently in $O(ABd^2)$ steps of computation [44, Lemma 3.9]. (2) Rather than estimating the reward at every $(a, b)$ in a tabular fashion, we use a weighted least-squares (9) to obtain estimates $(\widehat{\theta}_1, \widehat{\theta}_2)$ which are then used to approximate the true reward for all $(a, b)$.

We now state our main guarantee for Algorithm 3. The proof can be found in Appendix E.1.

**Theorem 5** (Learning Stackelberg in linear bandit games). *For any linear bandit game, Algorithm 3 outputs a $(\mathtt{gap}_\varepsilon + \varepsilon)$-approximate Stackelberg equilibrium $(\widehat{a}, \widehat{b})$ with probability at least $1 - \delta$:*

$$\phi_0(\widehat{a}) \geq \phi_{\varepsilon/2}(\widehat{a}) \geq \max_{a \in \mathcal{A}} \phi_0(a) - \mathtt{gap}_\varepsilon - \varepsilon,$$

*in at most $n = \widetilde{O}(d^2/\varepsilon^2)$ queries.*

**Sample complexity; computation** Theorem 5 shows that Algorithm 3 achieves $\widetilde{O}(d^2/\varepsilon^2)$ sample complexity for learning Stackelberg equilibria in linear bandit games. This only depends polynomially on the feature dimension $d$ instead of the size of the action spaces $A, B$, which improves over Algorithm 1 when $A, B$ are large and is desired given the linear structure (8). This sample complexity has at most a $\widetilde{O}(d)$ gap from the lower bound: An $\Omega(d/\varepsilon^2)$ lower bound for linear bandit games can be obtained by directly adapting $\Omega(AB/\varepsilon^2)$ lower bound for bandit games in Theorem 3 (see Appendix E.2 for a formal statement and proof). We also note that, while the focus of Theorem 5 is on the sample complexity rather than the computation, Algorithm 3 is guaranteed to run in $\mathrm{poly}(A, B, d, 1/\varepsilon^2)$ time, since the CoreSet subroutine, the weighted least squares step (9), and the final optimization step in approximate best-response sets can all be solved in polynomial time.

## 6 Conclusion

This paper provides the first line of sample complexity results for learning Stackelberg equilibria in general-sum games with bandit feedback of the rewards and sampling noise. We identify a fundamental gap between the exact and estimated versions of the Stackelberg value, and design sample-efficient algorithms for learning Stackelberg with value optimal up to this gap, in several representative two-player general-sum games. We believe our results open up a number of interesting future directions, such as the optimal sample complexity for bandit-RL games and linear-bandit games, learning Stackelberg in more general Markov games, or further characterizations on what kinds of games admit a small gap.

## Acknowledgment

The authors would like to thank Alex Trott and Stephan Zheng for the inspiring discussions on the AI Economist. The authors also thank the Theory of Reinforcement Learning program at the Simons Institute (Fall 2020) for hosting the authors and incubating our initial discussions.

## Funding transparency statement

YB, HW, CX are funded through employment with Salesforce.

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
