## A   Results with optimistic tie-breaking

In this section, we present alternative versions of our main results where the Stackelberg equilibrium is defined via *optimistic tie-breaking*.

### A.1   Bandit games

The setting is exactly the same as in Section 3, except that now we consider optimistic versions of the $\phi$-functions that take the **max** over best-response sets (henceforth the $\psi$-functions):

$$\psi_\varepsilon(a) := \max_{b \in \mathsf{BR}_\varepsilon(a)} \mu_1(a, b), \tag{11}$$

for all $\varepsilon \geq 0$. Notice that now $\psi_\varepsilon \geq \psi_0$, and we consider the following new definition of gap:

$$\widetilde{\mathsf{gap}}_\varepsilon := \max_{a \in \mathcal{A}_\varepsilon} [\psi_\varepsilon(a) - \psi_0(a)], \quad \text{where}$$

$$\mathcal{A}_\varepsilon := \left\{ a \in \mathcal{A} : \psi_\varepsilon(a) \geq \max_{a \in \mathcal{A}} \psi_0(a) - \varepsilon \right\}.$$

Our desired optimality guarantee is

$$\psi_0(\widehat{a}) \geq \max_{a \in \mathcal{A}} \psi_0(a) - \widetilde{\mathsf{gap}}_\varepsilon - \varepsilon.$$

We now state our sample complexity upper bound for learning Stackelberg in bandit games under optimistic tie-breaking. The proof can be found in Section F.1.

**Theorem A.1** (Bandit games with optimistic tie-breaking). *For the two-player bandit game and any* $\varepsilon \in (0, 1)$, *Algorithm 4 outputs* $(\widehat{a}, \widehat{b})$ *such that with probability at least* $1 - \delta$,

$$\psi_0(\widehat{a}) \geq \max_{a \in \mathcal{A}} \psi_0(a) - \widetilde{\mathsf{gap}}_\varepsilon - \varepsilon,$$

$$\mu_2(\widehat{a}, \widehat{b}) \geq \max_{b' \in \mathcal{B}} \mu_2(\widehat{a}, b') - \varepsilon$$

*with* $n = \widetilde{O}(AB/\varepsilon^2)$ *samples, where* $\widetilde{O}(\cdot)$ *hides log factors. Further, the algorithm runs in* $O(n) = \widetilde{O}(AB/\varepsilon^2)$ *time.*

**Intuitions about new gap**   We provide some intuitions about why—in contrast to the $\mathsf{gap}_\varepsilon$ defined in Section 3—our sample complexity depends on this newly defined $\widetilde{\mathsf{gap}}_\varepsilon$ here. Observe that, $\widetilde{\mathsf{gap}}_\varepsilon$ measures the max gap between $\psi_\varepsilon(a) - \psi_0(a)$, over all possible $a$'s whose $\psi_\varepsilon$ can compete with the best $\psi_0$. For any of these $a$'s, statistically (if we only have $O(AB/\varepsilon^2)$ samples), the best response set $\mathsf{BR}_\varepsilon(a)$ is indistinguishable from $\mathsf{BR}_0(a)$, and we may well pick these $a$'s as the Stackelberg equilibrium. However, their true $\psi_0$ can be (much) lower than the $\psi_\varepsilon$, and thus picking one of these $a$'s we may have to suffer from the so-defined $\mathsf{gap}_\varepsilon$ in the worst case.

**Algorithm 5** Learning Stackelberg in bandit-RL games with optimistic tie-breaking
***

**Require:** Target accuracy $\varepsilon > 0$.

**set** $N \leftarrow \widetilde{O}(H^5 S^2 B/\varepsilon^2 + H^7 S^4 B/\varepsilon)$.

1: **for** $a \in \mathcal{A}$ **do**
2:      Let the leader pull arm $a \in \mathcal{A}$ and the follower run the `Reward-Free RL-Explore` algorithm for $N$ episodes, and obtain model estimate $\widehat{M}^a$.
3:      Let $(\widehat{V}_1(a, \cdot), \widehat{V}_2(a, \cdot))$ denote the value functions for the model $\widehat{M}^a$.
4:      Compute the empirical best response value $\widehat{V}_2^\star(a) := \max_{\pi^b} \widehat{V}_2(a, \pi^b)$ by any optimal planning algorithm (e.g. value iteration) on the empirical MDP $\widehat{M}^a$.
5:      Solve the following program

$$
\begin{aligned}
& \text{maximize}_{\pi^b} \;\; \widehat{V}_1(a, \pi^b) \\
& \text{s.t.} \;\; \pi^b \in \widehat{\mathsf{BR}}_{3\varepsilon/4}(a) := \left\{ \pi^b : \widehat{V}_2(a, \pi^b) \geq \widehat{V}_2^\star(a) - 3\varepsilon/4 \right\}
\end{aligned}
\tag{12}
$$

     by subroutine $(\widehat{\pi}^{b,(a)}, \widehat{\psi}_{3\varepsilon/4}(a)) \leftarrow \texttt{BestCaseBestResponse}(\widehat{M}^a, \widehat{V}_2^\star(a) - 3\varepsilon/4)$.

**output** $(\widehat{a}, \widehat{\pi}^b)$, where $\widehat{a} \leftarrow \arg\max_{a \in \mathcal{A}} \widehat{\psi}_{3\varepsilon/4}(a)$ and $\widehat{\pi}^b \leftarrow \widehat{\pi}^{b,(\widehat{a})}$.
***

## A.2 Bandit-RL games

The setting is exactly the same as in Section 4, except the definition of the $\psi$-functions takes the max over best-response sets:

$$
\psi_\varepsilon(a) := \max_{\pi^b \in \mathsf{BR}_\varepsilon(a)} V_1(a, \pi^b),
$$

for all $\varepsilon \geq 0$. Similar as in Section A.1, we consider the following new definition of gap:

$$
\widetilde{\mathsf{gap}}_\varepsilon := \max_{a \in \mathcal{A}_\varepsilon} [\psi_\varepsilon(a) - \psi_0(a)], \quad \text{where}
$$

$$
\mathcal{A}_\varepsilon := \left\{ a \in \mathcal{A} : \psi_\varepsilon(a) \geq \max_{a \in \mathcal{A}} \psi_0(a) - \varepsilon \right\}.
$$

We now state our sample complexity upper bound for learning Stackelberg in bandit-RL games under optimistic tie-breaking. The proof is analogous to Theorem 4, and can be found in Section F.2.

**Theorem A.2** (Learning Stackelberg in bandit-RL games with optimistic tie-breaking)**.** *For any bandit-RL game and sufficiently small $\varepsilon \leq O(1/H^2 S^2)$, Algorithm 5 with $n = \widetilde{O}(H^5 S^2 AB/\varepsilon^2 + H^7 S^4 AB/\varepsilon)$ episodes of play can return $(\widehat{a}, \widehat{\pi}^b)$ such that with probability at least $1 - \delta$,*

$$
\psi_0(\widehat{a}) \geq \max_{a \in \mathcal{A}} \psi_0(a) - \widetilde{\mathsf{gap}}_\varepsilon - \varepsilon,
$$

$$
V_2(\widehat{a}, \widehat{\pi}^b) \geq \max_{\widetilde{\pi}^b} V_2(\widehat{a}, \widetilde{\pi}^b) - \varepsilon,
$$

*where $\widetilde{O}(\cdot)$ hides $\log(HSAB/\delta\varepsilon)$ factors. Further, the algorithm runs in $\mathrm{poly}(HSAB/\delta\varepsilon)$ time.*

## B Matrix game with simultaneous play

In this section, we consider a variant of the two-player bandit game, in which the leader and follower instead play a matrix game simultaneously, and the follower cannot see the leader's action. The problem of finding Stackelberg in this setting is also known as learning the "optimal strategy to commit to" [13].

**Setting** A general-sum matrix game with simultaneous play can be described as $M = (\mathcal{A}, \mathcal{B}, r_1, r_2)$ with $|\mathcal{A}| = A$, $|\mathcal{B}| = B$, and $r_1, r_2 : \mathcal{A} \times \mathcal{B} \to [0, 1]$, which defines the following game:

- The leader pre-specifies a policy $\pi^a \in \Delta_{\mathcal{A}}$ and reveals this policy to the follower.

- The leader plays $a \sim \pi^a$, and the follower plays an action $b \in \mathcal{B}$ simultaneously without seeing $a$.

- The leader receives reward $r_1(a, b)$ and the follows receives reward $r_2(a, b)$.

(Above, $\Delta_{\mathcal{A}}$ denotes the probability simplex on $\mathcal{A}$.) Let $\mu_i(\pi^a, b) = \sum_{a' \in \mathcal{A}} \pi^a(a') \mathbb{E}[r_i(a', b)]$, $i = 1, 2$ denote the mean rewards (for mixed policies), and

$$\phi_\varepsilon(\pi^a) := \min_{b \in \mathsf{BR}_\varepsilon(\pi^a)} \mu_1(\pi^a, b),$$

$$\mathsf{BR}_\varepsilon(\pi^a) := \left\{ b \in \mathcal{B} : \mu_2(\pi^a, b) \geq \max_{b'} \mu_2(\pi^a, b') - \varepsilon \right\}$$

denote the $\varepsilon$-approximate best response sets and best response values for any $\varepsilon \geq 0$, similar as in bandit games. We also overload notation to let $\phi_\varepsilon(a_1) := \phi_\varepsilon(\delta_{a_1})$ to denote the $\phi_\varepsilon$ value at pure strategies ($\delta_a$ is the pure strategy of always taking $a$).

The main difference between this setting and bandit games is that now the Stackelberg equilibrium for the leader may be achieved at *mixed strategies* only, so that we can no longer restrict attention to pure strategies $a \in \mathcal{A}$ for the leader. To see why this is true, consider 2x2 game of [13] shown in Table 1. In this game, the two pure strategies $\{a_1, a_2\}$ achieve $\phi_0(a_1) = 2$ (since the best response is $b_1$) and $\phi_0(a_2) = 3$ (since the best response is $b_2$). However, if we take $\pi_p^a = p\delta_{a_1} + (1-p)\delta_{a_2}$, then the follower's best response is $b_2$ whenever $p < 1/2$. Taking $p \to (1/2)_-$, the leader can achieve value $\phi_0(\pi_p^a) = 4p + 3(1 - p) \to 3.5$, which is higher than both pure strategies. For $p \geq 1/2$, the follower's best response is $b_1$, and $\phi_0(\pi_p^a) \leq 2$. Therefore the Stackelberg equilibrium for the leader is to take $\pi_p^a$ with $p \to (1/2)_-$[4].

| $\mu_1, \mu_2$ | $b_1$ | $b_2$ |
|:---:|:---:|:---:|
| $a_1$ | $2, 1$ | $4, 0$ |
| $a_2$ | $1, 0$ | $3, 1$ |

Table 1: Example of matrix game with simultaneous play, where the Stackelberg strategy for the leader is mixed.

## B.1 Main result

Let $\mathsf{gap}_\varepsilon := \sup_{\pi^a \in \Delta_{\mathcal{A}}} \phi_0(\pi^a) - \sup_{\pi^a \in \Delta_{\mathcal{A}}} \phi_\varepsilon(\pi^a)$ denote the gap. The following result shows that $\widetilde{O}(AB/\varepsilon^2)$ samples suffice for learning the Stackelberg up to $(\mathsf{gap}_\varepsilon + \varepsilon)$ in simultaneous matrix games, similar as in bandit games. The proof can be found in Appendix G.1.

**Theorem B.1** (Learning Stackelberg in simultaneous matrix games). *For any matrix game with simultaneous play, Algorithm 6 queries for $n = O(AB \log(AB/\delta)/\varepsilon^2) = \widetilde{O}(AB/\varepsilon^2)$ samples, and outputs $(\widehat{\pi}^a, \widehat{b})$ such that with probability at least $1 - \delta$,*

$$\phi_0(\widehat{\pi}^a) \geq \phi_{\varepsilon/2}(\widehat{\pi}^a) \geq \sup_{\pi^a \in \Delta_{\mathcal{A}}} \phi_0(\pi^a) - \mathsf{gap}_\varepsilon - \varepsilon,$$

$$\mu_2(\widehat{\pi}^a, \widehat{b}) \geq \max_{b' \in \mathcal{B}} \mu_2(\widehat{\pi}^a, b') - \varepsilon.$$

Theorem B.1 implies that $\widetilde{O}(AB/\varepsilon^2)$ samples is also enough for determining the approximate (up to gap) Stackelberg equilibrium in simultaneous games. Also, as we assumed bandit feedback, Theorem B.1 extends the results of Letchford et al. [27], Peng et al. [34] which studied the sample complexity assuming a best response oracle (can query $\mathsf{BR}_0(\pi^a)$ for any $\pi^a \in \Delta_{\mathcal{A}}$).

**Comparison between learning Stackelberg and Nash** We compare Theorem B.1 with existing results on learning Nash equilibria in general-sum matrix games. On the one hand, when we have $\widetilde{O}(AB/\varepsilon^2)$ samples, with only a $(\mathsf{gap}_\varepsilon + \varepsilon)$ near-optimal Stackelberg equilibrium, but we can learn

---

[4]The reason why the optimal policy can only be approached instead of exactly achieved is because of the pessimistic tie-breaking at $p = 1/2$, and is resolved if we take optimistic tie-breaking.

---

**Algorithm 6** Learning Stackelberg in matrix games with simultaneous play

---

**Require:** Target accuracy $\varepsilon > 0$.

**set** $N \leftarrow C \log(AB/\delta)/\varepsilon^2$ for some constant $C > 0$.

1: Query each $(a, b) \in \mathcal{A} \times \mathcal{B}$ for $N$ times and obtain $\{r_1^{(j)}(a, b), r_2^{(j)}(a, b)\}_{j=1}^N$.

2: Construct empirical estimates $\widehat{\mu}_i(\pi^a, b) = \sum_{a' \in \mathcal{A}} \pi^a(a') \frac{1}{N} \sum_{j=1}^N r_i^{(j)}(a', b)$ for $i = 1, 2$.

3: Construct approximate best response sets and values for all $\pi^a \in \Delta_{\mathcal{A}}$:

$$\widehat{\mathsf{BR}}_{3\varepsilon/4}(\pi^a) := \left\{ b : \widehat{\mu}_2(\pi^a, b) \geq \max_{b' \in \mathcal{B}} \widehat{\mu}_2(\pi^a, b') - 3\varepsilon/4 \right\},$$

$$\widehat{\phi}_{3\varepsilon/4}(\pi^a) := \min_{b \in \widehat{\mathsf{BR}}_{3\varepsilon/4}(\pi^a)} \widehat{\mu}_1(\pi^a, b).$$

4: Output $(\widehat{\pi}^a, \widehat{b})$ such that

$$\widehat{\phi}_{3\varepsilon/4}(\widehat{\pi}^a) \geq \sup_{\pi^a \in \Delta_{\mathcal{A}}} \widehat{\phi}_{3\varepsilon/4}(\pi^a) - \varepsilon/2, \tag{13}$$

$$\widehat{b} = \operatorname*{arg\,min}_{b \in \widehat{\mathsf{BR}}_{3\varepsilon/4}(\widehat{\pi}^a)} \widehat{\mu}_1(\widehat{\pi}^a, b).$$

---

an $\varepsilon$-approximate Nash equilibrium [28]. On the other hand, the Stackelberg value is uniquely defined (as the max of $\phi_0$), whereas there can be multiple Nash values [38] for general-sum games. Additionally, at the Stackelberg equilibrium, the leader's payoff is guaranteed to be at least as good as any Nash value (the leader can pre-specify any Nash policy). This makes Stackelberg a perhaps better solution concept in asymmetric games where the learning goal focuses more on the leader.

**Runtime** In Algorithm 6, the step of approximately maximizing $\widehat{\phi}_{3\varepsilon/4}(\pi^a)$ in (13) requires optimizing a discontinuous function over a continuous domain. It is unclear whether this program can be reformulated to be solved efficiently in polynomial time[5]. However, we remark that this is special to the pessimistic tie-breaking we assumed [27]. Learning the Stackelberg equilibrium with optimistic tie-breaking has the same $\widetilde{O}(AB/\varepsilon^2)$ sample complexity while admitting an efficient polynomial-time algorithm via linear programming (see Section B.2 for the formal statement and proof).

## B.2 Optimistic tie-breaking

We also study simultaneous matrix games with optimistic tie-breaking. The setting is exactly the same as above except the definition of the $\psi$-functions takes the max over best-response sets:

$$\psi_\varepsilon(\pi^a) := \max_{b \in \mathsf{BR}_\varepsilon(\pi^a)} \mu_1(\pi^a, b),$$

for all $\varepsilon \geq 0$. Similar as in bandit games (Section A.1), we consider the following new definition of gap:

$$\widetilde{\mathsf{gap}}_\varepsilon := \max_{\pi^a \in \mathcal{A}_\varepsilon} [\psi_\varepsilon(\pi^a) - \psi_0(\pi^a)], \quad \text{where}$$

$$\mathcal{A}_\varepsilon := \left\{ \pi^a \in \Delta_{\mathcal{A}} : \psi_\varepsilon(\pi^a) \geq \max_{\pi^a \in \Delta_{\mathcal{A}}} \psi_0(\pi^a) - \varepsilon \right\}.$$

---

[5]This program has a finite-time solution by the following strategy (which utilizes the specific structure of this program): First partition $\Delta_{\mathcal{A}}$ according to which subsets of $\mathcal{B}$ are $3\varepsilon/4$ best response, and then within each partition solve an linear program (over $\pi^a$) to a fixed accuracy (e.g. $\varepsilon/10$). However, the runtime is exponential because there are $2^B$ subsets induced by the partition.

[6]We also remark that [47, Theorem 9 & Proposition 10] provides an efficient reformulation of the pessimistic Stackelberg problem in simultaneous matrix games. However, their reformulation relies crucially on the best response set being *exact*, and does not generalize to our setting which requires to solve the pessimistic Stackelberg problem with *approximate* best response sets.

**Algorithm 7** Learning Stackelberg in matrix games with simultaneous play (optimistic tie-breaking version)

---

**Require:** Target accuracy $\varepsilon > 0$.
**set** $N \leftarrow C \log(AB/\delta)/\varepsilon^2$ for some constant $C > 0$.

1: Query each $(a, b) \in \mathcal{A} \times \mathcal{B}$ for $N$ times and obtain $\{r_1^{(j)}(a, b), r_2^{(j)}(a, b)\}_{j=1}^N$.

2: Construct empirical estimates $\widehat{\mu}_i(\pi^a, b) = \sum_{a' \in \mathcal{A}} \pi^a(a') \frac{1}{N} \sum_{j=1}^N r_i^{(j)}(a', b)$ for $i = 1, 2$.

3: Construct approximate best response sets and values for all $\pi^a \in \Delta_{\mathcal{A}}$:

$$\widehat{\mathsf{BR}}_{3\varepsilon/4}(\pi^a) := \left\{ b : \widehat{\mu}_2(\pi^a, b) \geq \max_{b' \in \mathcal{B}} \widehat{\mu}_2(\pi^a, b') - 3\varepsilon/4 \right\},$$

$$\widehat{\phi}_{3\varepsilon/4}(\pi^a) := \max_{b \in \widehat{\mathsf{BR}}_{3\varepsilon/4}(\pi^a)} \widehat{\mu}_1(\pi^a, b).$$

4: Output

$$\widehat{\pi}^a = \arg\max_{\pi^a \in \Delta_{\mathcal{A}}} \widehat{\phi}_{3\varepsilon/4}(\pi^a), \tag{14}$$

$$\widehat{b} = \arg\max_{b \in \widehat{\mathsf{BR}}_{3\varepsilon/4}(\widehat{\pi}^a)} \widehat{\mu}_1(\widehat{\pi}^a, b).$$

By calling the subroutine $(\widehat{\pi}^a, \widehat{b}) \leftarrow \texttt{BestMixedLeaderStrategy}(\widehat{\mu}_1, \widehat{\mu}_2)$.

---

**Theorem B.2** (Learning Stackelberg in simultaneous matrix games with optimistic tie-breaking).
*For any matrix game with simultaneous play, Algorithm 7 queries for $n = O\big(AB \log(AB/\delta)/\varepsilon^2\big) = \widetilde{O}(AB/\varepsilon^2)$ samples, and outputs $(\widehat{\pi}^a, \widehat{b})$ such that with probability at least $1 - \delta$,*

$$\psi_0(\widehat{\pi}^a) \geq \max_{\pi^a \in \Delta_{\mathcal{A}}} \psi_0(\pi^a) - \widetilde{\mathsf{gap}}_\varepsilon - \varepsilon,$$

$$\mu_2(\widehat{\pi}^a, \widehat{b}) \geq \max_{b' \in \mathcal{B}} \mu_2(\widehat{\pi}^a, b') - \varepsilon.$$

*Further, the algorithm runs in* $\mathrm{poly}(n)$ *time.*

The proof can be found in Section G.2.

**Efficient runtime**   Theorem B.2 shares the same sample complexity $\widetilde{O}(AB/\varepsilon^2)$ as its pessimistic tie-breaking counterpart (Theorem B.1), albeit with a slightly different definition of the gap. However, an additional advantage of the optimistic version is that it is guaranteed to have a polynomial runtime.
The core reason behind this is that with optimistic tie-breaking now $(\widehat{\pi}^a, \widehat{b})$ solves a *max-max problem* (instead of a max-min problem), for which we can exchange the order of maximization. Concretely, we can now first maximize over $\pi^a$ for each $b$, which admits a linear programming formulation (cf. the `BestMixedLeaderStrategy` subroutine in Algorithm 10, also in [13]).

## C   Proofs for Section 3

### C.1   Proof of Theorem 1

To prove Theorem 1, we will construct a pair of hard instances, and use Le Cam's method [49, Section 15.2] to reduce the estimation error into a testing problem between the two hard instances. Consider the following two games $M_1$ and $M_{-1}$, where the rewards follow Bernoulli distributions: $r_i(a, b) \sim \mathsf{Ber}(\mu_i(a, b))$ with means shown in Table 2, where $\delta \in (0, 1)$ is a parameter to be determined:

Based on Table 2, it is straightforward to check that $\phi_0^{M_1}(a_1) = 1$, $\phi_0^{M_{-1}}(a_1) = 0$, and $\phi_0^{M_1}(a_2) = \phi_0^{M_{-1}}(a_2) = 1/2$. Further, $a_\star^{M_1} = a_1$ and $a_\star^{M_{-1}} = a_2$.

For any algorithm that outputs a (possibly randomized) estimator $\widehat{a} \in \mathcal{A}$ of the Stackelberg equilibrium, let $\pi$ denotes its querying policy, that is, given prior queries and observations

| $r_1, r_2$ | $b_1$ | $b_2$ |
|---|---|---|
| $a_1$ | $1, \frac{1+\delta}{2}$ | $0, \frac{1-\delta}{2}$ |
| $a_2$ | $\frac{1}{2}, 1$ | $\frac{1}{2}, 1$ |

| $r_1, r_2$ | $b_1$ | $b_2$ |
|---|---|---|
| $a_1$ | $1, \frac{1-\delta}{2}$ | $0, \frac{1+\delta}{2}$ |
| $a_2$ | $\frac{1}{2}, 1$ | $\frac{1}{2}, 1$ |

Table 2: Pair of hard instances $M_1$ (left) and $M_{-1}$ (right). Each table lists $\mu_1(a,b), \mu_2(a,b)$ for $a \in \{a_1, a_2\}, b \in \{b_1, b_2\}$.

$\left\{ a^{(i)}, b^{(i)}, r_1^{(i)}, r_2^{(i)} \right\}_{i=1}^{k-1}$, $\pi^{(k)}(a, b| \left\{ a^{(i)}, b^{(i)}, r_1^{(i)}, r_2^{(i)} \right\}_{i=1}^{k-1})$ denotes the distribution of the next query. Let $\mathbb{P}_{M_1, \pi}$ and $\mathbb{P}_{M_{-1}, \pi}$ denote the distribution of all $n$ observations generated by the querying policy $\pi$. For these two instances, we have

$$\sup_{M \in \{M_1, M_{-1}\}} \mathbb{P}_M \left( \max_{a \in \mathcal{A}} \phi_0^M(a) - \phi_0^M(\widehat{a}) \geq \frac{1}{2} \right)$$

$$= \sup_{M \in \{M_1, M_{-1}\}} \mathbb{P}_M \left( \widehat{a} \neq \arg\max_{a \in \mathcal{A}} \phi_0^M(a) \right)$$

$$\geq \frac{1}{2} \left( \mathbb{P}_{M_1}(\widehat{a} \neq a_1) + \mathbb{P}_{M_{-1}}(\widehat{a} \neq a_2) \right)$$

$$\geq \frac{1}{2} \left( 1 - \text{TV}(\mathbb{P}_{M_1, \pi}, \mathbb{P}_{M_{-1}, \pi}) \right)$$

$$\geq \frac{1}{2} \left( 1 - \sqrt{\frac{1}{2} \text{KL}(\mathbb{P}_{M_1, \pi} \| \mathbb{P}_{M_{-1}, \pi})} \right),$$

where the second-to-last step used Le Cam's inequality, and the last step used Pinsker's inequality. To upper bound the KL distance between $\mathbb{P}_{M_1, \pi}$ and $\mathbb{P}_{M_{-1}, \pi}$, we apply the divergence decomposition of [23, Lemma 15.1] and obtain that

$$\text{KL}(\mathbb{P}_{M_1, \pi} \| \mathbb{P}_{M_{-1}, \pi}) \leq \sum_{(a,b) \in \mathcal{A} \times \mathcal{B}} \mathbb{E}_{M_1, \pi}[T_{a,b}(n)] \cdot \text{KL}\left( \mathbb{P}_{M_1}^{a,b} \| \mathbb{P}_{M_{-1}}^{a,b} \right) \leq n \cdot \max_{(a,b) \in \mathcal{A} \times \mathcal{B}} \text{KL}\left( \mathbb{P}_{M_1}^{a,b} \| \mathbb{P}_{M_{-1}}^{a,b} \right),$$

where $T_{a,b}(n)$ denotes the number of queries to $(a, b)$ among the $n$ queries, and $\mathbb{P}_{M_i}^{a,b}$ denote the distribution of the observation $(r_1(a, b), r_2(a, b))$ in problem $M_i$, $i = 1, 2$. We have $\text{KL}(\mathbb{P}_{M_1}^{a,b} \| \mathbb{P}_{M_{-1}}^{a,b}) = 0$ for $(a, b) = (a_2, b_1)$ and $(a, b) = (a_2, b_2)$ since these $(a, b)$ yield exactly the same reward distributions. For $(a, b) = (a_1, b_1)$ and $(a, b) = (a_1, b_2)$, using the bound $\text{KL}(\text{Ber}(\frac{1+\delta}{2}) \| \text{Ber}(\frac{1-\delta}{2})) = \delta \log \frac{1+\delta}{1-\delta} \leq 3\delta^2$ for $\delta \leq 1/2$ (and the same bound for $\text{KL}(\text{Ber}(\frac{1-\delta}{2}) \| \text{Ber}(\frac{1+\delta}{2}))$). Therefore, we get

$$\text{KL}(\mathbb{P}_{M_1, \pi} \| \mathbb{P}_{M_{-1}, \pi}) \leq 3n\delta^2.$$

Choosing $\delta = 1/\sqrt{(27/2)n}$, the above is upper bounded by $2/9$, and thus plugging back to the preceding bound yields

$$\sup_{M \in \{M_1, M_{-1}\}} \mathbb{P}\left( \widehat{a} \neq \arg\max_{a \in \mathcal{A}} \phi_0^M(a) \right) \geq \frac{1}{2} \left( 1 - \sqrt{\frac{1}{2} \text{KL}(\mathbb{P}_{M_1, \pi} \| \mathbb{P}_{M_{-1}, \pi})} \right) \geq \frac{1}{3}.$$

Therefore, choosing the problem class to be $\mathcal{M}_n = \{M_1, M_{-1}\}$ with $\delta = 1/\sqrt{(27/2)n}$, the above is the desired lower bound. $\qquad\square$

### C.2 A Lemma on the gap

**Lemma C.1** (Gap can be $\Omega(1)$). *For any $0 \leq \varepsilon_1 < \varepsilon_2 < 1$, there exists a two-player bandit game $M = M_{\varepsilon_1, \varepsilon_2}$ with $A = B = 2$, such that*

$$\max_{a \in \mathcal{A}} \phi_{\varepsilon_1}(a) - \max_{a \in \mathcal{A}} \phi_{\varepsilon_2}(a) \geq \frac{1}{2},$$

$$\max_{a \in \mathcal{A}} \phi_{\varepsilon_1}(a) - \phi_{\varepsilon_1}\left( \arg\max_{a' \in \mathcal{A}} \phi_{\varepsilon_2}(a') \right) \geq \frac{1}{2}.$$

*In particular, (taking $\varepsilon_1 = 0$), for any $\varepsilon$ there exists a game in which $\mathsf{gap}_\varepsilon = \max_{a \in \mathcal{A}} \phi_0(a) - \max_{a \in \mathcal{A}} \phi_\varepsilon(a) \geq 1/2$.*

*Proof.* Let $0 \leq \varepsilon_1 < \varepsilon_2$. We construct the problem $M = M_{\varepsilon_1,\varepsilon_2}$ as follows: $\mathcal{A} = \{a_1, a_2\}$ and $\mathcal{B} = \{b_1, b_2\}$, and the rewards $\{r_1(a,b), r_2(a,b)\}_{a \in \mathcal{A}, b \in \mathcal{B}}$ are all deterministic and valued as in the following table:

| $r_1, r_2$ | $b_1$ | $b_2$ |
|---|---|---|
| $a_1$ | $1, \frac{\varepsilon_1 + \varepsilon_2}{2}$ | $0, 0$ |
| $a_2$ | $\frac{1}{2}, 1$ | $\frac{1}{2}, 1$ |

Table 3: Construction of $r_1(a,b), r_2(a,b)$ for $a \in \{a_1, a_2\}, b \in \{b_1, b_2\}$.

For the arm $a_2$, actions $b_1$ and $b_2$ are exactly the same, so we have $\phi_\varepsilon(a_2) = \frac{1}{2}$ for all $\varepsilon$. For the arm $a_1$, observe that $\varepsilon_1 < \frac{\varepsilon_1 + \varepsilon_2}{2} < \varepsilon_2$, and thus $\mathsf{BR}_{\varepsilon_1}(a_1) = \{b_1\}$ and $\phi_{\varepsilon_1}(a_1) = 1$, but $\mathsf{BR}_{\varepsilon_2}(a_1) = \{b_1, b_2\}$ and $\phi_{\varepsilon_2}(a_1) = 0$. Therefore,

$$\max_{a \in \mathcal{A}} \phi_{\varepsilon_1}(a) = \max \left\{ 1, \frac{1}{2} \right\} = 1,$$

$$\max_{a \in \mathcal{A}} \phi_{\varepsilon_2}(a) = \max \left\{ 0, \frac{1}{2} \right\} = \frac{1}{2},$$

$$\phi_{\varepsilon_1} \left( \arg\max_{a' \in \mathcal{A}} \phi_{\varepsilon_2}(a') \right) = \phi_{\varepsilon_1}(a_2) = \frac{1}{2}.$$

This shows the desired result. $\qquad\square$

### C.3 Proof of Theorem 2

Algorithm 1 pulled each arm $(a,b)$ for $N = O(\log(AB/\delta)/\varepsilon^2)$ times, and $\widehat{\mu}_1(a,b), \widehat{\mu}_2(a,b)$ are the empirical means of the observed rewards. By the Hoeffding inequality and union bound over $(a,b)$, with probability at least $1 - \delta$, we have

$$\max_{(a,b) \in \mathcal{A} \times \mathcal{B}} |\widehat{\mu}_i(a,b) - \mu_i(a,b)| \leq \varepsilon/8 \quad \text{for } i = 1, 2. \tag{15}$$

**Properties of $\widehat{\mathsf{BR}}_{3\varepsilon/4}(a)$**    On the uniform convergence event (22), we have the following: for any $b \in \mathsf{BR}_{\varepsilon/2}(a)$, we have

$$\widehat{\mu}_2(a,b) \geq \mu_2(a,b) - \varepsilon/8 \geq \max_{b' \in \mathcal{B}} \mu_2(a,b') - 5\varepsilon/8 \geq \max_{b' \in \mathcal{B}} \widehat{\mu}_2(a,b') - 3\varepsilon/4,$$

and thus $b \in \widehat{\mathsf{BR}}_{3\varepsilon/4}(a)$. This shows that $\mathsf{BR}_{\varepsilon/2}(a) \subseteq \widehat{\mathsf{BR}}_{3\varepsilon/4}(a)$. Similarly we can show that $\widehat{\mathsf{BR}}_{3\varepsilon/4}(a) \subseteq \mathsf{BR}_\varepsilon(a)$. In other words,

$$\mathsf{BR}_\varepsilon(a) \supseteq \widehat{\mathsf{BR}}_{3\varepsilon/4}(a) \supseteq \mathsf{BR}_{\varepsilon/2}(a) \quad \text{for all } a \in \mathcal{A}.$$

Notably, this implies that $\widehat{b} \in \widehat{\mathsf{BR}}_{3\varepsilon/4}(\widehat{a}) \in \mathsf{BR}_\varepsilon(\widehat{a})$, the desired near-optimality guarantee for $\widehat{b}$.

**Near-optimality of $\widehat{a}$**    On the one hand, because $\widehat{a}$ maximizes $\widehat{\mu}_1(a, \widehat{b}(a))$, we have for any $a \in \mathcal{A}$ that

$$\min_{b' \in \widehat{\mathsf{BR}}_{3\varepsilon/4}(\widehat{a})} \widehat{\mu}_1(\widehat{a}, b') \overset{(i)}{=} \widehat{\mu}_1(\widehat{a}, \widehat{b}) \geq \widehat{\mu}_1(a, \widehat{b}(a)) \overset{(ii)}{\geq} \min_{b \in \mathsf{BR}_\varepsilon(a)} \widehat{\mu}_1(a, b),$$

where (i) is because $\widehat{b}$ minimizes $\widehat{\mu}_1(\widehat{a}, \cdot)$ within $\widehat{\mathsf{BR}}_{3\varepsilon/4}(\widehat{a})$, and (ii) is because $\widehat{b}(a) \in \widehat{\mathsf{BR}}_{3\varepsilon/4}(a) \subseteq \mathsf{BR}_\varepsilon(a)$. By the uniform convergence (22), we get that

$$\min_{b' \in \widehat{\mathsf{BR}}_{3\varepsilon/4}(\widehat{a})} \mu_1(\widehat{a}, b') \geq \min_{b \in \mathsf{BR}_\varepsilon(a)} \mu_1(a,b) - 2 \cdot \varepsilon/8 \geq \phi_\varepsilon(a) - \varepsilon.$$

Since the above holds for all $a \in \mathcal{A}$, taking the max on the right hand side gives

$$\max_{a \in \mathcal{A}} \phi_\varepsilon(a) - \varepsilon \leq \min_{b' \in \widehat{\mathsf{BR}}_{3\varepsilon/4}(\widehat{a})} \mu_1(\widehat{a}, b') \overset{(i)}{\leq} \min_{b' \in \mathsf{BR}_{\varepsilon/2}(\widehat{a})} \mu_1(\widehat{a}, b') = \phi_{\varepsilon/2}(\widehat{a}),$$

where (i) is because $\widehat{\mathsf{BR}}_{3\varepsilon/4}(\widehat{a}) \supseteq \mathsf{BR}_{\varepsilon/2}(a)$. This yields that

$$\phi_{\varepsilon/2}(\widehat{a}) \geq \max_{a \in \mathcal{A}} \phi_\varepsilon(a) - \varepsilon = \max_{a \in \mathcal{A}} \phi_0(a) - \mathsf{gap}_\varepsilon - \varepsilon,$$

which is the first part of the bound for $\widehat{a}$.

On the other hand, since $\phi_\varepsilon(a)$ is increasing as we decrease $\varepsilon$, we directly have

$$\phi_{\varepsilon/2}(\widehat{a}) \leq \phi_0(\widehat{a}) \leq \max_{a \in \mathcal{A}} \phi_0(a).$$

This is the second part of the bound for $\widehat{a}$. $\qquad\square$

### C.4   Proof of Theorem 3

Suppose $\varepsilon \in (0, c)$ and $g \in [0, c)$ where $c > 0$ is an absolute constant to be determined. For any algorithm that outputs an estimator $\widehat{a} \in \mathcal{A}$, let $\pi$ denote its (sequential) querying policy, and $\mathbb{P}_{M,\pi}$ denote the joint distribution of the $N$ observed rewards $(r_1(a^{(i)}, b^{(i)}), r_2(a^{(i)}, b^{(i)}))_{i=1}^N$ under game $M$. We will rely on the divergence decomposition of [23, Lemma 15.1]:

$$\mathrm{KL}(\mathbb{P}_{M,\pi} \| \mathbb{P}_{M',\pi}) \leq \sum_{(a,b) \in \mathcal{A} \times \mathcal{B}} \mathbb{E}_{M_1,\pi}[T_{a,b}(N)] \cdot \mathrm{KL}\left(\mathbb{P}_M^{a,b} \| \mathbb{P}_{M'}^{a,b}\right), \tag{16}$$

where $P_M^{a,b}$ denotes the distribution of observation $(r_1(a, b), r_2(a, b))$ under game $M$, and $T_{a,b}(N)$ denotes the number of queries of $(a, b)$ using algorithm $\pi$ (which is a random variable). We will also use the fact that

$$\mathrm{KL}(\mathsf{Ber}(1/2) \| \mathsf{Ber}(1/2 + \delta)) = \frac{1}{2} \log \frac{1/2}{1/2 + \delta} + \frac{1}{2} \log \frac{1/2}{1/2 - \delta} = \frac{1}{2} \log \frac{1}{1 - 4\delta^2} \leq \frac{1}{2} \cdot 8\delta^2 \leq 4\delta^2 \tag{17}$$

whenever $4\delta^2 \leq 1/2$, i.e. $|\delta| \leq 1/2\sqrt{2}$.

**Construction of hard instance**    In our construction below, the rewards follow Bernoulli distributions: $r_i(a, b) \sim \mathsf{Ber}(\mu_i(a, b))$, so that it suffices to specify $\mu_i(a, b)$. Without loss of generality assume $B/3$ is an integer, and let $\mathcal{B} = [B] = \{1, \ldots, B\}$ for notational simplicity.

We define a family of games $M_{a_\star, b_\star^1, b_\star^2}$ indexed by $a_\star \in \mathcal{A}$ and $b_\star^1, b_\star^2 \in \mathcal{B}$. Each game $M_{a_\star, b_\star^1, b_\star^2}$ is defined as follows:

- $\mu_1(a, b) = 1/2 + g + \varepsilon$ for all $a \in \mathcal{A}$ and $1 \leq b \leq B/3$;
- $\mu_1(a, b) = 1/2 + \varepsilon$ for all $a \in \mathcal{A}$ and $B/3 + 1 \leq b \leq 2B/3$;
- $\mu_1(a, b) = 1/2$ for all $a \in \mathcal{A}$ and $2B/3 + 1 \leq b \leq B$.
- $\mu_2(a_\star, b_\star^1) = 1/2 + 2\varepsilon$, where $b_\star^1 \in \{1, \ldots, B/3\}$.
- $\mu_2(a_\star, b_\star^2) = 1/2 + 5\varepsilon/4$, where $b_\star^2 \in \{B/3 + 1, \ldots, 2B/3\}$.
- $\mu_2(a_\star, b') = 1/2$ for all $b' \neq b_\star^1, b_\star^2$.
- $\mu_2(a', b) = 1/2$ for all $a' \neq a_\star$ and $b \in \mathcal{B}$.

For this game, we have $\phi_0(a_\star) = 1/2 + g + \varepsilon$, $\phi_\varepsilon(a_\star) = 1/2 + \varepsilon$, and $\phi_0(a') = \phi_\varepsilon(a') = 1/2$ for all $a' \neq a_\star$. Therefore,

$$\mathsf{gap}_\varepsilon = \max_{a \in \mathcal{A}} \phi_0(a) - \max_{a \in \mathcal{A}} \phi_\varepsilon(a) = g.$$

Further, notice that as long as $\widehat{a} \neq a_\star$, we have $\phi_0(\widehat{a}) = \max_a \phi_0(a) - (g + \varepsilon)$.

Define $\mathbb{P}_M$ as the mixture of over the prior of $M_{a_\star, b_\star^1, b_\star^2}$ where the prior samples $a_\star \sim \mathsf{Unif}(\mathcal{A})$, $b_\star^1 \sim \mathsf{Unif}(\{1, \ldots, B/3\})$, and $b_\star^2 \sim \mathsf{Unif}(\{B/3 + 1, \ldots, 2B/3\})$. Define $M_0$ as the "null-game" where all the $r_2$ are $1/2$, and $r_1$ has the same configuration as in the above game.

**Proof of lower bound**   Under the mixture $\mathbb{P}_M$, we have

$$\mathbb{P}_M\left(\phi_0(\widehat{a}) \leq \max_{a \in \mathcal{A}} \phi_0(a) - (g + \varepsilon)\right) = \mathbb{P}_M(\widehat{a} \neq a_\star)$$

$$= \frac{1}{A(B^2/9)} \sum_{a_\star} \sum_{b_\star^1=1}^{B/3} \sum_{b_\star^2=B/3+1}^{2B/3} \mathbb{P}_{M_{a_\star,b_\star^1,b_\star^2}}(\widehat{a} \neq a_\star)$$

$$\geq \frac{1}{A(B^2/9)} \sum_{a_\star} \sum_{b_\star^1=1}^{B/3} \sum_{b_\star^2=B/3+1}^{2B/3} \mathbb{P}_0(\widehat{a} \neq a_\star) - \frac{1}{A(B^2/9)} \sum_{a_\star} \sum_{b_\star^1=1}^{B/3} \sum_{b_\star^2=B/3+1}^{2B/3} \mathrm{TV}\left(\mathbb{P}_{0,\pi}, \mathbb{P}_{M_{a_\star,b_\star^1,b_\star^2},\pi}\right)$$

$$\geq \frac{1}{A} \sum_{a_\star} \mathbb{P}_0(\widehat{a} \neq a_\star) - \frac{1}{A(B^2/9)} \sum_{a_\star} \sum_{b_\star^1=1}^{B/3} \sum_{b_\star^2=B/3+1}^{2B/3} \sqrt{\frac{1}{2}\mathrm{KL}\left(\mathbb{P}_{0,\pi}\|\mathbb{P}_{M_{a_\star,b_\star^1,b_\star^2},\pi}\right)}$$

$$\geq 1 - \frac{1}{A} - \underbrace{\frac{1}{A(B^2/9)} \sum_{a_\star} \sum_{b_\star^1=1}^{B/3} \sum_{b_\star^2=B/3+1}^{2B/3} \sqrt{\frac{1}{2}\mathrm{KL}\left(\mathbb{P}_{0,\pi}\|\mathbb{P}_{M_{a_\star,b_\star^1,b_\star^2},\pi}\right)}}_{(\star)} .$$

We now show that $(\star) \leq 1/3$ if $N \leq c[AB/\varepsilon^2]$ for some small absolute constant $c > 0$. Using the divergence decomposition (16), we have

$$(\star) \leq \frac{1}{A(B^2/9)} \sum_{a_\star} \sum_{b_\star^1=1}^{B/3} \sum_{b_\star^2=B/3+1}^{2B/3} \sqrt{\frac{1}{2} \sum_{a,b} \mathbb{E}_{0,\pi}[T_{a,b}(N)]\mathrm{KL}\left(\mathbb{P}_0^{a,b}\|\mathbb{P}_{M_{a_\star,b_\star^1,b_\star^2}}^{a,b}\right)}$$

$$\stackrel{(i)}{=} \frac{1}{A(B^2/9)} \sum_{a_\star} \sum_{b_\star^1=1}^{B/3} \sum_{b_\star^2=B/3+1}^{2B/3} \left(\frac{1}{2}\mathbb{E}_{0,\pi}\left[T_{a_\star,b_\star^1}(N)\right]\mathrm{KL}\left(\mathbb{P}_0^{a_\star,b_\star^1}\|\mathbb{P}_{M_{a_\star,b_\star^1,b_\star^2}}^{a_\star,b_\star^1}\right) + \right.$$

$$\left. \frac{1}{2}\mathbb{E}_{0,\pi}\left[T_{a_\star,b_\star^2}(N)\right]\mathrm{KL}\left(\mathbb{P}_0^{a_\star,b_\star^2}\|\mathbb{P}_{M_{a_\star,b_\star^1,b_\star^2}}^{a_\star,b_\star^2}\right)\right)^{1/2}$$

$$\leq \frac{1}{A(B^2/9)} \sum_{a_\star} \sum_{b_\star^1=1}^{B/3} \sum_{b_\star^2=B/3+1}^{2B/3} \sqrt{\frac{1}{2}\mathbb{E}_{0,\pi}\left[T_{a_\star,b_\star^1}(N)\right]\mathrm{KL}\left(\mathbb{P}_0^{a_\star,b_\star^1}\|\mathbb{P}_{M_{a_\star,b_\star^1,b_\star^2}}^{a_\star,b_\star^1}\right)}$$

$$+ \sqrt{\frac{1}{2}\mathbb{E}_{0,\pi}\left[T_{a_\star,b_\star^2}(N)\right]\mathrm{KL}\left(\mathbb{P}_0^{a_\star,b_\star^2}\|\mathbb{P}_{M_{a_\star,b_\star^1,b_\star^2}}^{a_\star,b_\star^2}\right)}$$

$$\stackrel{(ii)}{\leq} \frac{1}{A(B/3)} \sum_{a_\star} \sum_{b_\star^1=1}^{B/3} \sqrt{\frac{1}{2}\mathbb{E}_{0,\pi}\left[T_{a_\star,b_\star^1}(N)\right] \cdot 4 \cdot (2\varepsilon)^2} + \frac{1}{A(B/3)} \sum_{a_\star} \sum_{b_\star^2=1}^{B/3} \sqrt{\frac{1}{2}\mathbb{E}_{0,\pi}\left[T_{a_\star,b_\star^2}(N)\right] \cdot 4 \cdot (5\varepsilon/4)^2}$$

$$\stackrel{(iii)}{\leq} \sqrt{\frac{1}{A(B/3)} \sum_{a_\star} \sum_{b_\star^1=1}^{B} \mathbb{E}_{0,\pi}\left[T_{a_\star,b_\star^1}(N)\right] \cdot 8\varepsilon^2} + \sqrt{\frac{1}{A(B/3)} \sum_{a_\star} \sum_{b_\star^2=1}^{B} \mathbb{E}_{0,\pi}\left[T_{a_\star,b_\star^2}(N)\right] \cdot 8\varepsilon^2}$$

$$= 2\sqrt{\frac{24N\varepsilon^2}{AB}} .$$

Above, (i) used the fact that for the null game $M_0$ and the game $M_{a_\star,b_\star^1,b_\star^2}$, only the actions $(a_\star, b_\star^1)$ and $(a_\star, b_\star^2)$ will lead to different observation distributions. (ii) used the fact that $r_2(a_\star, b_\star^1)$ has mean $1/2 + 2\varepsilon$ under $M_{a_\star,b_\star^1,b_\star^2}$ and mean $1/2$ under $M_0$ (and the other Bernoulli means correspondingly), and the fact that $r_1(a, b)$ are equally distributed in the two games and thus do not contribute to the KL, and finally the KL bound (17) for small enough $\varepsilon$ such that $2\varepsilon < 1/2\sqrt{2}$. (iii) used the power mean inequality and the equality $\sum_{a_\star,b_\star} T_{a_\star,b_\star}(N) = N$ for any algorithm.

The above implies that, as long as $\varepsilon < 1/4\sqrt{2}$ and $g \leq 1/4$, for $N \leq AB/(864\varepsilon^2)$, we have $(\star) \leq 1/3$, and thus

$$\mathbb{P}_M\left(\phi_0(\widehat{a}) \leq \max_{a \in \mathcal{A}} \phi_0(a) - (g + \varepsilon)\right) = \frac{1}{A(B^2/9)} \sum_{a_\star, b_\star^1, b_\star^2} \mathbb{P}_{M_{a_\star, b_\star^1, b_\star^2}}\left(\phi_0(\widehat{a}) \leq \max_{a \in \mathcal{A}} \phi_0(a) - (g + \varepsilon)\right)$$

$$\geq 1 - \frac{1}{A} - \frac{1}{3} \geq \frac{1}{3}.$$

Therefore there must exist a game $M_{a_\star, b_\star^1, b_\star^2}$ on which the error probability is at least $1/3$. This is the desired lower bound. $\qquad\square$

## C.5 Equivalence to turn-based Markov game

We consider the following general-sum turn-based Markov game[7] with two steps and state space $\mathcal{S} = \{s_a : a \in \mathcal{A}\}$:

- ($h = 1$) Leader receives deterministic initial state $s_1$ and plays action $a \in \mathcal{A}$. No reward for both players.
- ($h = 2$) The game transits deterministically to $s_a$. The follower plays action $b \in \mathcal{B}$ and observes reward $r_2(s_a, b) = r_2(a, b)$. The leader observes reward $r_1(s_a, b) = r_1(a, b)$.
- The game terminates.

It is straightforward to see that the bandit game $M = (\mathcal{A}, \mathcal{B}, r_1, r_2)$ is equivalent to the above turn-based Markov game. Note that the Markov game has $|\mathcal{S}| = A$ states.

Now, let $a_\star$ be the leader's exact Stackelberg equilibrium (as defined in (3)). For any $a$, let $b_\star(a) = \arg\min_{b \in \mathsf{BR}_0(a)} \mu_1(a, b)$ be the best response of $a$ with the worst $\mu_1$. Define the deterministic follower policy $\pi_\star^b$ as $\pi_\star^b(s_a) = b_\star(a)$ for all $s_a \in \mathcal{S}$.

We claim that $(a_\star, \pi_\star^b)$ is a Nash equilibrium of the above Markov game. Indeed, $\pi_\star^b$ is clearly $a_\star$'s best response on the follower's reward. Also, if we fix $\pi_\star^b$, then $a_\star$ is also the leader's best response to $\pi_\star^b$, as we have

$$\mu_1(s_a, \pi_\star^b(s_a)) = \mu_1(a, b_\star(a)) = \min_{b \in \mathsf{BR}_0(a)} \mu_1(a, b) = \phi_0(a),$$

and thus the leader's best response is exactly the argmax of $\phi_0(a)$, i.e. $a_\star$.

## C.6 Additional discussions on the gap

Here we show that for bandit games, $\mathsf{gap}_\varepsilon$ is small for two special kinds of games: zero-sum games and cooperative games.

**Zero-sum games** Here $r_1 \equiv -r_2$ and thus $\mu_1 \equiv -\mu_2$. In such games, by definition we have

$$\phi_\varepsilon(a) = \min_{b \in \mathsf{BR}_\varepsilon(a)} \mu_1(a, b),$$

$$\mathsf{BR}_\varepsilon(a) = \left\{b \in \mathcal{B} : \mu_1(a, b) \leq \min_{b'} \mu_1(a, b') + \varepsilon\right\}.$$

Notice that now the minimum over $b \in \mathsf{BR}_\varepsilon(a)$ is always taken at the exact minimizer of $b \mapsto \mu_1(a, b)$. Therefore we have $\phi_\varepsilon(a) = \min_{b \in \mathcal{B}} \mu_1(a, b)$ does not depend on $\varepsilon$, and thus $\mathsf{gap}_\varepsilon = \max_{a \in \mathcal{A}} \phi_0(a) - \max_{a \in \mathcal{A}} \phi_\varepsilon(a) = 0$.

**Cooperative games** Here $r_1 \equiv r_2$ and thus $\mu_1 \equiv \mu_2$. In such games, by definition we have

$$\phi_\varepsilon(a) = \min_{b \in \mathsf{BR}_\varepsilon(a)} \mu_1(a, b),$$

$$\mathsf{BR}_\varepsilon(a) = \left\{b \in \mathcal{B} : \mu_1(a, b) \geq \max_{b'} \mu_1(a, b') - \varepsilon\right\}.$$

Thus for each $a \in \mathcal{A}$, the difference between $\mu_1(a, b)$ for any $b \in \mathsf{BR}_0(a)$ and any $b \in \mathsf{BR}_\varepsilon(a)$ is at most $\varepsilon$. This shows that $\phi_0(a) - \phi_\varepsilon(a) \leq \varepsilon$ for all $a \in \mathcal{A}$ and thus $\mathsf{gap}_\varepsilon \leq \varepsilon$.

---

[7]The formal definition of turn-based Markov games can be found in [4].

# D  Proofs for Section 4

## D.1  Subroutine `WorstCaseBestResponse`

We describe the `WorstCaseBestResponse` subroutine in Algorithm 8.

---

**Algorithm 8** Subroutine `WorstCaseBestResponse`$(M, \underline{V}_2)$

---
**Require:** MDP $M = (H, \mathcal{S}, \mathcal{B}, \mathbb{P}_h(\cdot|\cdot, \cdot), r_{1,h}(\cdot, \cdot), r_{2,h}(\cdot, \cdot))$. Initial state $s_1 \in \mathcal{S}$. Target value $\underline{V}_2$.
 1: Solve the following linear program over $\{d_h(s, b) : h \in [H], s \in \mathcal{S}, b \in \mathcal{B}\}$:

$$\text{minimize} \sum_{h=1}^{H} \sum_{s \in \mathcal{S}, b \in \mathcal{B}} d_h(s, b) r_{1,h}(s, b)$$

$$\text{s.t.} \sum_{h=1}^{H} \sum_{s \in \mathcal{S}, b \in \mathcal{B}} d_h(s, b) r_{2,h}(s, b) \geq \underline{V}_2, \tag{18}$$

$$\sum_{s \in \mathcal{S}, b \in \mathcal{B}} d_h(s, b) \mathbb{P}_h(s'|s, b) = \sum_{b' \in \mathcal{B}} d_{h+1}(s', b') \quad \text{for all } 1 \leq h \leq H - 1, \ s' \in \mathcal{S},$$

$$d_1(s_1, \cdot) \in \Delta_{\mathcal{B}}, \quad d_1(s'_1, \cdot) = 0 \quad \text{for all } s' \neq s_1.$$

Above, $\Delta_{\mathcal{B}}$ denotes the probability simplex on $\mathcal{B}$ (which is a set of $B + 1$ linear constraints). Let $d_h$ denote the solution and $\underline{V}_1$ denote the value of the above program.
 2: Set $\pi_h^b(b|s) \leftarrow d_h(s, b)/\sum_{b \in \mathcal{B}} d_h(s, b)$ for all $(h, s, b)$ (with the convention $0/0 = 1/B$).
**output** $(\pi^b, \underline{V}_1)$.

---

## D.2  Proof of Theorem 4

**Correctness of subroutine**  We first show that the `WorstCaseBestResponse` subroutine (Algorithm 8) with input $(\widehat{M}^a, \widehat{V}_2^\star(a) - 3\varepsilon/4)$ indeed solves the nominal problem (7). To see this, observe that in the nominal problem (7), both the objective function and the constraint are linear functions of the visitation distribution $\left\{ \mathbb{P}_h^{\pi^b}(s, b) \right\}$ induced by $\pi^b$. Therefore, maximizing over all visitation distributions is equivalent to maximizing over all $\pi^b$. To ensure that a general $\{d_h(s, b)\}_{h,s,b}$ is a visitation distribution, it suffices for it to satisfy the constraints $d_1(s_1, \cdot) \in \Delta_{\mathcal{B}}, d_1(s'_1, \cdot) = 0$ for $s'_1 \neq s_1$, and at each $h \geq 2$ and each state $s'$ the in-flow is equal to the out-flow, meaning that

$$\sum_{s \in \mathcal{S}, b \in \mathcal{B}} d_h(s, b) \mathbb{P}_h(s'|s, b) = \sum_{b \in \mathcal{B}} d_{h+1}(s', b)$$

for all $h \geq 1$, $s' \in \mathcal{S}$. These are exactly the constraints specified in the linear program (18). Finally, for a visitation distribution $d_h$, notice that $\pi_h^b(b|s) = d_h(s, b)/\sum_{b \in \mathcal{S}} d_h(s, b)$ (with the convention $0/0 = 1/B$) defines a policy $\pi^b$ whose visitation distribution is exactly $d_h$. This shows that the linear program (18) is indeed a correct algorithm for solving (7).

**Properties of reward-free exploration**  For each $a \in \mathcal{A}$, Algorithm 2 played the `Reward-Free RL-Explore` algorithm of Jin et al. [18] for $N = \widetilde{O}(H^5 S^2 B/\varepsilon^2 + H^7 S^4 B/\varepsilon)$ episodes and obtained an estimate of the transition dynamics $\widehat{\mathbb{P}}^a$. (More specifically, it ran $N_0 = \widetilde{O}(H^7 S^4 B/\varepsilon)$ episodes in its *exploration* phase and $N_{\text{data}} = \widetilde{O}(H^5 S^2 B/\varepsilon^2)$ episodes in its *data-gathering* phase.) Further, let $\{\widehat{r}_{1,h}(a, s, b), \widehat{r}_{2,h}(a, s, b)\}$ denote the empirical mean of the observed rewards in the data-gathering phase.

Let $\widetilde{V}_1(a, \pi^b)$ and $\widetilde{V}_2(a, \pi^b)$ denote the value functions of the empirical MDPs $(\mathbb{P}^a, \widehat{r}_1)$ and $(\mathbb{P}^a, \widehat{r}_2)$ (note that these MDPs combine the true models and the *empirical* rewards). With our choice $N$, by [18, Theorem 3.1], we have with probability at least $1 - \delta$ that

$$\sup_{\pi^b} \left| \widehat{V}_i(a, \pi^b) - \widetilde{V}_i(a, \pi^b) \right| \leq \varepsilon/16 \quad \text{for } i = 1, 2. \tag{19}$$

We now argue that the `Reward-Free RL-Explore` algorithm can correctly estimate the rewards, along with estimating transitions. Indeed, we have the following

**Lemma D.1.** *Suppose we run the `Reward-Free RL-Explore` algorithm where the data gathering phase contains $N_{\mathrm{data}} \geq \widetilde{O}(H^3 S^2 B/\varepsilon^2)$ trajectories, and we in addition receive (stochastic) reward signals $r_{1,h}, r_{2,h}$ along the trajectories. Then with probability at least $1 - \delta$, the empirical reward estimates $\widehat{r}_{1,h}, \widehat{r}_{2,h}$ and the associated value functions $\widetilde{V}_1$ and $\widetilde{V}_2$ satisfy that*

$$\sup_{\pi^b} \left| \widetilde{V}_i(a, \pi^b) - V_i(a, \pi^b) \right| \leq \varepsilon \quad \text{for } i = 1, 2.$$

We defer the proof of Lemma D.1 to Appendix D.3. As we have $N_{\mathrm{data}} = \widetilde{O}(H^5 S^2 B/\varepsilon^2)$, we can apply Lemma 6 and get (by choosing a large absolute constant in the choice of $N_{\mathrm{data}}$) that

$$\sup_{\pi^b} \left| \widetilde{V}_i(a, \pi^b) - V_i(a, \pi^b) \right| \leq \varepsilon/16 \quad \text{for } i = 1, 2. \tag{20}$$

Combining (19) and (20) (and noticing those are true for all $a \in \mathcal{A}$), we get

$$\sup_{a \in \mathcal{A}, \pi^b} \left| \widehat{V}_i(a, \pi^b) - V_i(a, \pi^b) \right| \leq \varepsilon/8 \quad \text{for } i = 1, 2. \tag{21}$$

**Guarantees on $\mathsf{BR}_{3\varepsilon/4}(a)$** Now, for any $a \in \mathcal{A}$, recall Algorithm 2 constructed the empirical best-response set (cf. (7))

$$\widehat{\mathsf{BR}}_{3\varepsilon/4}(a) := \left\{ \pi^b : \widehat{V}_2(a, \pi^b) \geq \max_{\widetilde{\pi}^b} \widehat{V}_2(a, \widetilde{\pi}^b) - 3\varepsilon/4 \right\}.$$

We claim that

$$\mathsf{BR}_\varepsilon(a) \supseteq \widehat{\mathsf{BR}}_{3\varepsilon/4}(a) \supseteq \mathsf{BR}_{\varepsilon/2}(a) \quad \text{for all } a \in \mathcal{A}.$$

Indeed, fixing any $a \in \mathcal{A}$, let $\pi^\star$ denote the optimal policy for $V_2(a, \cdot)$ and $\widehat{\pi}^\star$ denote the optimal policy for $\widehat{V}_2(a, \cdot)$ (dropping dependence on $a$ for notational simplicity). Suppose $\pi'_b \in \widehat{\mathsf{BR}}_{3\varepsilon/4}(a)$, then we have

$$V_2(a, \pi^\star) - V_2(a, \pi'_b)$$
$$\leq \underbrace{V_2(a, \pi^\star) - \widehat{V}_2(a, \pi^\star)}_{\leq \varepsilon/8} + \underbrace{\widehat{V}_2(a, \pi^\star) - \widehat{V}_2(a, \widehat{\pi}^\star)}_{\leq 0} + \underbrace{\widehat{V}_2(a, \widehat{\pi}^\star) - \widehat{V}_2(a, \pi'_b)}_{\leq 3\varepsilon/4} + \underbrace{\widehat{V}_2(a, \pi'_b) - V_2(a, \pi'_b)}_{\leq \varepsilon/8}$$
$$\leq 3\varepsilon/4 + 2 \cdot \varepsilon/8 \leq \varepsilon.$$

This shows that $\widehat{\mathsf{BR}}_{3\varepsilon/4}(a) \subseteq \mathsf{BR}_\varepsilon(a)$. Notably, this implies that the output $\widehat{\pi}^b \in \mathsf{BR}_\varepsilon(\widehat{a})$, the desired optimality guarantee for $\widehat{\pi}^b$.

Similar as above, take any $\pi'_b \in \mathsf{BR}_{\varepsilon/2}(a)$, we have

$$\widehat{V}_2(a, \widehat{\pi}^\star) - \widehat{V}_2(a, \pi'_b)$$
$$\leq \underbrace{\widehat{V}_2(a, \widehat{\pi}^\star) - V_2(a, \widehat{\pi}^\star)}_{\leq \varepsilon/8} + \underbrace{V_2(a, \widehat{\pi}^\star) - V_2(a, \pi^\star)}_{\leq 0} + \underbrace{V_2(a, \pi^\star) - V_2(a, \pi'_b)}_{\leq \varepsilon/2} + \underbrace{V_2(a, \pi'_b) - \widehat{V}_2(a, \pi'_b)}_{\leq \varepsilon/8}$$
$$\leq \varepsilon/2 + 2 \cdot \varepsilon/8 \leq 3\varepsilon/4.$$

This shows that $\mathsf{BR}_{\varepsilon/2}(a) \subseteq \widehat{\mathsf{BR}}_{3\varepsilon/4}(a)$, the other part of the claim.

**Stackelberg guarantee for $\widehat{a}$** Finally, we show the Stackelberg guarantee for $\widehat{a}$. This part is similar as in the proof of Theorem 2. First, because $\widehat{a}$ maximizes $\widehat{\phi}_{3\varepsilon/4}(a) = \min_{\pi^b \in \widehat{\mathsf{BR}}_{3\varepsilon/4}(a)} \widehat{V}_1(a, \pi^b)$, we have for any $a \in \mathcal{A}$ that

$$\min_{\widetilde{\pi}^b \in \widehat{\mathsf{BR}}_{3\varepsilon/4}(\widehat{a})} \widehat{V}_1(\widehat{a}, \widetilde{\pi}^b) = \widehat{\phi}_{3\varepsilon/4}(\widehat{a}) \geq \widehat{\phi}_{3\varepsilon/4}(a) = \min_{\widetilde{\pi}^b \in \widehat{\mathsf{BR}}_{3\varepsilon/4}(a)} \widehat{V}_1(a, \widetilde{\pi}^b) \overset{(i)}{\geq} \min_{\widetilde{\pi}^b \in \mathsf{BR}_\varepsilon(a)} \widehat{V}_1(a, \widetilde{\pi}^b)$$

where (i) is because $\widehat{\mathsf{BR}}_{3\varepsilon/4}(a) \subseteq \mathsf{BR}_\varepsilon(a)$ for all $a$. By the uniform convergence (21), we get

$$\min_{\widetilde{\pi}^b \in \widehat{\mathsf{BR}}_{3\varepsilon/4}(\widehat{a})} V_1(\widehat{a}, \widetilde{\pi}^b) \geq \min_{\widetilde{\pi}^b \in \mathsf{BR}_\varepsilon(a)} V_1(a, \widetilde{\pi}^b) - 2 \cdot \varepsilon/8 \geq \phi_\varepsilon(a) - \varepsilon.$$

Since the above holds for all $a \in \mathcal{A}$, taking the max on the right hand side gives

$$\max_{a \in \mathcal{A}} \phi_\varepsilon(a) - \varepsilon \leq \min_{\widetilde{\pi}^b \in \widehat{\mathsf{BR}}_{3\varepsilon/4}(\widehat{a})} V_1(\widehat{a}, \widetilde{\pi}^b) \overset{(i)}{\leq} \min_{\widetilde{\pi}^b \in \mathsf{BR}_{\varepsilon/2}(\widehat{a})} V_1(\widehat{a}, \widetilde{\pi}^b) = \phi_{\varepsilon/2}(\widehat{a}),$$

where (i) is because $\widehat{\mathsf{BR}}_{3\varepsilon/4}(\widehat{a}) \supseteq \mathsf{BR}_{\varepsilon/2}(a)$. In other words, we have

$$\phi_{\varepsilon/2}(\widehat{a}) \geq \max_{a \in \mathcal{A}} \phi_\varepsilon(a) - \varepsilon = \max_{a \in \mathcal{A}} \phi_0(a) - \mathsf{gap}_\varepsilon - \varepsilon.$$

This is the first part of the bound for $\widehat{a}$.

On the other hand, since $\phi_\varepsilon(a)$ is increasing as we decrease $\varepsilon$, we directly have

$$\phi_{\varepsilon/2}(\widehat{a}) \leq \phi_0(\widehat{a}) \leq \max_{a \in \mathcal{A}} \phi_0(a).$$

This is the second part of the bound for $\widehat{a}$.

$\square$

## D.3 Proof of Lemma D.1

We consider estimating a single reward $r_h = r_{1,h}$. The bound for two rewards can be obtained by setting $\delta \to \delta/2$ and applying a union bound. Consider the MDP $M^a$ which consists of $S$ states, $B$ actions, and $H$ steps. Let $V(a, \pi^b; r)$ denote the value function using the true MDP, policy $\pi^b$ and reward function $r$, and $V(a, \pi^b; \widehat{r})$ denote the value function using the estimated reward $\widehat{r}$. Further, let

$$\mathcal{S}_h^\delta := \left\{ s : \max_{\pi^b} \mathbb{P}_h^{\pi^b}(s) \geq \delta \right\}.$$

denote the set of $\delta$-significant states. By [18], the data gathering phase of `Reward-Free RL-Explore` obtains data where the $h$-th step is sampled i.i.d. from some policy $\mu_h$, such that for any $s \in \mathcal{S}_h^\delta$ we have

$$\max_{\pi^b} \frac{\mathbb{P}_h^{\pi^b}(s, b)}{\mu_h(s, b)} \leq 2SBH.$$

We have for any $\pi^b$ that

$$\left| \widetilde{V}_1(a, \pi^b) - V_1(a, \pi^b) \right| = \left| V(a, \pi^b; r) - V(a, \pi^b; \widehat{r}) \right|$$

$$= \left| \sum_{h=1}^H \sum_{s,b} \mathbb{P}_h^{\pi^b}(s, b)(\widehat{r}_h(s, b) - \mathbb{E}[r_h(s, b)]) \right|$$

$$\leq \left| \sum_{h=1}^H \sum_{s \notin \mathcal{S}_h^\delta, b} \mathbb{P}_h^{\pi^b}(s, b)(\widehat{r}_h(s, b) - \mathbb{E}[r_h(s, b)]) \right| + \left| \sum_{h=1}^H \sum_{s \in \mathcal{S}_h^\delta, a} \mathbb{P}_h^{\pi^b}(s, b)(\widehat{r}_h(s, b) - \mathbb{E}[r_h(s, b)]) \right|$$

$$\leq \sum_{h=1}^H \sum_{s \notin \mathcal{S}_h^\delta} \mathbb{P}_h^{\pi^b}(s) + \sum_{h=1}^H \underbrace{\left| \sum_{s \in \mathcal{S}_h^\delta, b} \mathbb{P}_h^{\pi^b}(s, b)(\widehat{r}_h(s, b) - \mathbb{E}[r_h(s, b)]) \right|}_{:= \Delta_h}$$

$$\leq HS\delta + \sum_{h=1}^H \Delta_h.$$

For any $h$, by the Cauchy-Schwarz inequality, we have

$$
\sup_{\pi^b} \Delta_h \leq \sup_{\pi^b} \left[ \sum_{s \in \mathcal{S}_h^\delta, b} \underbrace{\mathbb{P}_h^{\pi^b}(s, b)}_{=\mathbb{P}_h^{\pi^b}(s) \cdot \pi_h^b(b|s)} (\widehat{r}_h(s, b) - \mathbb{E}[r_h(s, b)])^2 \right]^{1/2}
$$

$$
\leq \sup_{\pi^b} \max_{\nu: \mathcal{S} \to \mathcal{B}} \left[ \sum_{s \in \mathcal{S}_h^\delta, b} \mathbb{P}_h^{\pi^b}(s)(\widehat{r}_h(s, b) - \mathbb{E}[r_h(s, b)])^2 \mathbf{1}\{b = \nu(s)\} \right]^{1/2}
$$

$$
\overset{(i)}{\leq} \max_{\nu: \mathcal{S} \to \mathcal{B}} \left[ 2SBH \cdot \sum_{s \in \mathcal{S}_h^\delta, b} \mu_h(s, b)(\widehat{r}_h(s, b) - \mathbb{E}[r_h(s, b)])^2 \mathbf{1}\{b = \nu(s)\} \right]^{1/2}
$$

$$
\overset{(ii)}{\leq} \max_{\nu: \mathcal{S} \to \mathcal{B}} \left[ 2SBH \cdot \sum_{s \in \mathcal{S}_h^\delta, b} \mu_h(s, b) \cdot \widetilde{O}\left( \frac{1}{N_h(s, b)} \right) \cdot \mathbf{1}\{b = \nu(s)\} \right]^{1/2}
$$

$$
\overset{(iii)}{\leq} \max_{\nu: \mathcal{S} \to \mathcal{B}} \left[ 2SBH \cdot \sum_{s \in \mathcal{S}_h^\delta, b} \mu_h(s, b) \cdot \widetilde{O}\left( \frac{1}{N \mu_h(s, b)} \right) \cdot \mathbf{1}\{b = \nu(s)\} \right]^{1/2}
$$

$$
= \widetilde{O}\left( \sqrt{\frac{S^2 BH}{N}} \right).
$$

Above, (i) used the fact that $\mathbb{P}_h^{\pi^b}(s)\mathbf{1}\{b = \nu(s)\} \leq 2SBH \cdot \mu_h(s, b)$ as $\{\pi_{h'}^b\}_{h' \leq h-1} \cup \{\nu\}$ is a valid policy. (ii) used the Hoeffding inequality (and a union bound) for the reward estimates, and the fact that the visitation of the reward-free algorithm is independent of the observed reward. (iii) used the multiplicative Chernoff bound for the visitation count $N_h(s, b) \sim \text{Bin}(N, \mu_h(s, b))$ and a union bound over $(s, b)$, which requires $N \geq O(1/\min_{s,b} \mu_h(s, b))$. Recall that `Reward-Free RL-Explore` used $\delta = \varepsilon/2H^2 S$ and $\mu_h(s, b) \geq \frac{\varepsilon}{4H^3 S^2 B}$ for all $(s, b)$. Thus the requirement for $N$ is $N \geq O(H^3 S^2 B/\varepsilon)$ which is implied by our assumption that $N \geq \widetilde{O}(H^3 S^2 B/\varepsilon^2)$.

Further, plugging in the choice of $N$ (with a sufficiently large constant) into the above bound yields

$$
\sup_{\pi^b} \Delta_h \leq \widetilde{O}\left( \frac{S^2 BH}{H^3 S^2 B/\varepsilon^2} \right) \leq \varepsilon/2H.
$$

This further implies that

$$
\left| \widetilde{V}_1(a, \pi^b) - V_1(a, \pi^b) \right| \leq HS \cdot \varepsilon/(2H^2 S) + H \cdot \varepsilon/(2H) \leq \varepsilon,
$$

the desired result. $\qquad\square$

# E   Proofs for Section 5

## E.1   Proof of Theorem 5

First by the guarantee (10) for the `CoreSet` subroutine, we have $K = |\mathcal{K}| \leq 4d \log \log d + 16$. At each $j \in [K]$ and associated $(a_j, b_j) \in \mathcal{K}$, as we queried the rewards for $N$ times, the empirical means satisfy (letting $\phi_j := \phi(a_j, b_j)$ for shorthand)

$$
\widehat{\mu}_{i,j} = \phi_j^\top \theta_i^\star + \widetilde{z}_{i,j}, \quad i = 1, 2,
$$

where $\widetilde{z}_{i,j}$ is the empirical mean of $N$ i.i.d. 1-sub-Gaussian noises, and thus is $1/N$-sub-Gaussian. Therefore, the weighted least squares estimator (9) can be expressed as (letting $\rho_j := \rho(a_j, b_j)$ for shorthand)

$$
\widehat{\theta}_i = \arg\min_{\theta \in \mathbb{R}^d} \sum_{i=1}^K \rho(a_j, b_j)\big(\phi(a_j, b_j)^\top \theta - \widehat{\mu}_{i,j}\big)^2 = \left( \sum_{j=1}^K \rho_j \phi_j \phi_j^\top \right)^{-1} \sum_{j=1}^K \rho_j \phi_j \cdot \widehat{\mu}_{i,j}
$$

$$= \left(\sum_{j=1}^{K} \rho_j \phi_j \phi_j^\top\right)^{-1} \sum_{j=1}^{K} \rho_j \phi_j \left(\phi_j^\top \theta_i^\star + \widetilde{z}_{i,j}\right)$$

$$= \theta_i^\star + \left(\sum_{j=1}^{K} \rho_j \phi_j \phi_j^\top\right)^{-1} \sum_{j=1}^{K} \rho_j \phi_j \widetilde{z}_{i,j}.$$

This implies the following guarantee (recall $\Phi = \{\phi(a,b) : (a,b) \in \mathcal{A} \times \mathcal{B}\}$):

$$\max_{\phi \in \Phi} \left|\phi^\top (\widehat{\theta}_i - \theta_i^\star)\right|$$

$$= \max_{\phi \in \Phi} \left|\phi^\top \left(\sum_{j=1}^{K} \rho_j \phi_j \phi_j^\top\right)^{-1} \sum_{j=1}^{K} \rho_j \phi_j \widetilde{z}_{i,j}\right|$$

$$\leq \max_j |\widetilde{z}_{i,j}| \cdot \max_{\phi \in \Phi} \left|\sum_{j=1}^{K} \rho_j \phi^\top \left(\sum_{j=1}^{K} \rho_j \phi_j \phi_j^\top\right)^{-1} \phi_j\right|$$

$$\overset{(i)}{\leq} \max_j |\widetilde{z}_{i,j}| \cdot \max_{\phi \in \Phi} \left(\sum_{j=1}^{K} \rho_j \phi^\top \left(\sum_{j=1}^{K} \rho_j \phi_j \phi_j^\top\right)^{-1} \phi_j \phi_j^\top \left(\sum_{j=1}^{K} \rho_j \phi_j \phi_j^\top\right)^{-1} \phi\right)^{1/2}$$

$$= \max_j |\widetilde{z}_{i,j}| \cdot \max_{\phi \in \Phi} \left(\phi^\top \left(\sum_{j=1}^{K} \rho_j \phi_j \phi_j^\top\right)^{-1} \phi\right)^{1/2}$$

$$\overset{(ii)}{\leq} \sqrt{2d} \cdot \max_j |\widetilde{z}_{i,j}|.$$

Above, (i) uses Jensen's inequality (over the distribution induced by $\rho_j$), and (ii) used the property (10) of the core set. Now, as $\widetilde{z}_{i,j}$ is $1/N$-sub-Gaussian, with probability at least $1 - \delta$, we have

$$\max_{i=1,2} \max_{j \in [K]} |\widetilde{z}_{i,j}| \leq \sqrt{\frac{\log(2K/\delta)}{N}} \leq C\sqrt{\frac{\log(d/\delta)}{N}}.$$

Substituting this into the preceding bound yields

$$\max_{\phi \in \Phi} \left|\phi^\top (\widehat{\theta}_i - \theta_i^\star)\right| \leq C\sqrt{\frac{d \log(d/\delta)}{N}}.$$

Choosing $N = Cd \log(d/\delta)/\varepsilon^2$ guarantees that $\max_{\phi \in \Phi} \left|\phi^\top (\widehat{\theta}_i - \theta_i^\star)\right| \leq \varepsilon/8$. When this happens, we have for any $(a,b) \in \mathcal{A} \times \mathcal{B}$ and any $i = 1, 2$ that the estimated mean reward is close to the true reward:

$$\left|\phi(a,b)^\top \widehat{\theta}_i - \phi(a,b)^\top \theta_i^\star\right| \leq \varepsilon/8.$$

We can then proceed analogously to the proof of Theorem 2 to conclude that the output $(\widehat{a}, \widehat{b})$ satisfies $\phi_0(\widehat{a}) \geq \phi_{\varepsilon/2}(\widehat{a}) \geq \max_{a \in \mathcal{A}} \phi_0(a) - \mathsf{gap}_\varepsilon - \varepsilon$ and $\widehat{b} \in \mathsf{BR}_\varepsilon(\widehat{a})$. Further, notice that the total amount of queries is

$$NK \leq Cd \log(d/\delta)/\varepsilon^2 \cdot d \log \log d = \widetilde{O}(d^2/\varepsilon^2).$$

This proves Theorem 5. $\qquad\square$

## E.2 Lower bound

We present a $\Omega(d/\varepsilon^2)$ lower bound for linear bandit games. This shows that the sample complexity upper bound in our Theorem 5 has at most an $\widetilde{O}(d)$ factor from the lower bound.

**Theorem E.1** (Lower bound for linear bandit games). *There exists an absolute constant $c > 0$ such that the following holds. For any $\varepsilon \in (0, c)$, $g \in [0, c)$, and any algorithm that queries $n \leq c[d/\varepsilon^2]$ samples and outputs an estimate $\widehat{a} \in \mathcal{A}$, there exists a linear bandit game $M$ with feature dimension $d$, on which $\mathsf{gap}_\varepsilon = g$ and the algorithm suffers from the $(g + \varepsilon)$ error:*

$$\phi_{\varepsilon/2}(\widehat{a}) \leq \phi_0(\widehat{a}) \leq \max_{a \in \mathcal{A}} \phi_0(a) - g - \varepsilon.$$

*Proof.* This lower bound is a direct corollary of the $\Omega(AB/\varepsilon^2)$ lower bound in Theorem 3. Specifically, we can pick the size of the action spaces $A', B'$ so that $d/2 \leq A'B' \leq d$, and take $\phi(a, b) = \mathbf{1}_{a,b} \in \mathbb{R}^d$ where $\mathbf{1}_{a,b}$ is the standard basis vector with one at index $(a, b)$ (this index is understood as an index in $[d]$). This family of linear bandit games is equivalent to the family of bandit games with $A'B' \geq d/2$, for which any algorithm has to suffer from at least $(g + \varepsilon_\varepsilon)$ error if the number of queries $n \leq cd/\varepsilon^2 \leq cA'B'/\varepsilon^2$ by (the hard instance construction of) Theorem 3. This proves Theorem E.1. $\qquad\square$

# F   Proofs for Section A

## F.1   Proof of Theorem A.1

Recall that Algorithm 4 pulled each arm $(a, b)$ for $N = O(\log(AB/\delta)/\varepsilon^2)$ times, and $\widehat{\mu}_1(a, b)$, $\widehat{\mu}_2(a, b)$ are the empirical means of the observed rewards. By the Hoeffding inequality and union bound over $(a, b)$, with probability at least $1 - \delta$, we have

$$\max_{(a,b) \in \mathcal{A} \times \mathcal{B}} |\widehat{\mu}_i(a, b) - \mu_i(a, b)| \leq \varepsilon/8 \quad \text{for } i = 1, 2. \tag{22}$$

**Properties of $\widehat{\mathsf{BR}}_{3\varepsilon/4}(a)$**   On the uniform convergence event (22), we have the following: for any $b \in \mathsf{BR}_{\varepsilon/2}(a)$, we have

$$\widehat{\mu}_2(a, b) \geq \mu_2(a, b) - \varepsilon/8 \geq \max_{b' \in \mathcal{B}} \mu_2(a, b') - 5\varepsilon/8 \geq \max_{b' \in \mathcal{B}} \widehat{\mu}_2(a, b') - 3\varepsilon/4,$$

and thus $b \in \widehat{\mathsf{BR}}_{3\varepsilon/4}(a)$. This shows that $\mathsf{BR}_{\varepsilon/2}(a) \subseteq \widehat{\mathsf{BR}}_{3\varepsilon/4}(a)$. Similarly we can show that $\widehat{\mathsf{BR}}_{3\varepsilon/4}(a) \subseteq \mathsf{BR}_\varepsilon(a)$. In other words,

$$\mathsf{BR}_\varepsilon(a) \supseteq \widehat{\mathsf{BR}}_{3\varepsilon/4}(a) \supseteq \mathsf{BR}_{\varepsilon/2}(a) \quad \text{for all } a \in \mathcal{A}.$$

Notably, this implies that $\widehat{b} \in \widehat{\mathsf{BR}}_{3\varepsilon/4}(\widehat{a}) \in \mathsf{BR}_\varepsilon(\widehat{a})$, the desired near-optimality guarantee for $\widehat{b}$.

**Near-optimality of $\widehat{a}$**   On the one hand, because $\widehat{a}$ maximizes $\widehat{\mu}_1(a, \widehat{b}(a))$, we have for any $a \in \mathcal{A}$ that

$$\max_{b' \in \widehat{\mathsf{BR}}_{3\varepsilon/4}(\widehat{a})} \widehat{\mu}_1(\widehat{a}, b') \overset{(i)}{=} \widehat{\psi}_{3\varepsilon/4}(\widehat{a}) \geq \widehat{\psi}_{3\varepsilon/4}(a) \overset{(ii)}{\geq} \max_{b \in \mathsf{BR}_{\varepsilon/2}(a)} \widehat{\mu}_1(a, b),$$

where (i) is by definition of $\widehat{\psi}_{3\varepsilon/4}$, and (ii) is because $\widehat{\mathsf{BR}}_{3\varepsilon/4}(a) \supseteq \mathsf{BR}_\varepsilon(a)$. By the uniform convergence (22), we get that

$$\max_{b' \in \widehat{\mathsf{BR}}_{3\varepsilon/4}(\widehat{a})} \mu_1(\widehat{a}, b') \geq \max_{b \in \mathsf{BR}_{\varepsilon/2}(a)} \mu_1(a, b) - 2 \cdot \varepsilon/8 \geq \psi_{\varepsilon/2}(a) - \varepsilon.$$

Since the above holds for all $a \in \mathcal{A}$, taking the max on the right hand side gives

$$\max_{a \in \mathcal{A}} \psi_{\varepsilon/2}(a) - \varepsilon \leq \max_{b' \in \widehat{\mathsf{BR}}_{3\varepsilon/4}(\widehat{a})} \mu_1(\widehat{a}, b') \overset{(i)}{\leq} \max_{b' \in \mathsf{BR}_\varepsilon(\widehat{a})} \mu_1(\widehat{a}, b') = \psi_\varepsilon(\widehat{a}),$$

where (i) is because $\widehat{\mathsf{BR}}_{3\varepsilon/4}(\widehat{a}) \subseteq \mathsf{BR}_\varepsilon(\widehat{a})$. This yields that

$$\psi_\varepsilon(\widehat{a}) \geq \max_{a \in \mathcal{A}} \psi_{\varepsilon/2}(a) - \varepsilon \geq \max_{a \in \mathcal{A}} \psi_0(a) - \varepsilon,$$

and thus $\widehat{a} \in \mathcal{A}_\varepsilon$, and we can further rewrite the above as

$$\psi_0(\widehat{a}) \geq \max_{a \in \mathcal{A}} \psi_0(a) - \varepsilon - [\psi_0(\widehat{a}) - \psi_\varepsilon(\widehat{a})] \geq \max_{a \in \mathcal{A}} \psi_0(a) - \widetilde{\mathsf{gap}}_\varepsilon - \varepsilon.$$

which is the desired bound for $\widehat{a}$. $\qquad\square$

**Algorithm 9** Subroutine `BestCaseBestResponse`$(M, \underline{V}_2)$

**Require:** MDP $M = (H, \mathcal{S}, \mathcal{B}, \mathbb{P}_h(\cdot|\cdot, \cdot), r_{1,h}(\cdot, \cdot), r_{2,h}(\cdot, \cdot))$. Initial state $s_1 \in \mathcal{S}$. Target value $\underline{V}_2$.
  1: Solve the following linear program over $\{d_h(s, b) : h \in [H], s \in \mathcal{S}, b \in \mathcal{B}\}$:

$$\text{maximize } \sum_{h=1}^{H} \sum_{s \in \mathcal{S}, b \in \mathcal{B}} d_h(s, b) r_{1,h}(s, b)$$

$$\text{s.t. } \sum_{h=1}^{H} \sum_{s \in \mathcal{S}, b \in \mathcal{B}} d_h(s, b) r_{2,h}(s, b) \geq \underline{V}_2, \tag{23}$$

$$\sum_{s \in \mathcal{S}, b \in \mathcal{B}} d_h(s, b) \mathbb{P}_h(s'|s, b) = \sum_{b' \in \mathcal{B}} d_{h+1}(s', b') \quad \text{for all } 1 \leq h \leq H - 1, \ s' \in \mathcal{S},$$

$$d_1(s_1, \cdot) \in \Delta_{\mathcal{B}}, \quad d_1(s_1', \cdot) = 0 \quad \text{for all } s' \neq s_1.$$

Above, $\Delta_{\mathcal{B}}$ denotes the probability simplex on $\mathcal{B}$ (which is a set of $B + 1$ linear constraints).
Let $d_h$ denote the solution and $\underline{V}_1$ denote the value of the above program.
  2: Set $\pi_h^b(b|s) \leftarrow d_h(s, b) / \sum_{b \in \mathcal{B}} d_h(s, b)$ for all $(h, s, b)$ (with the convention $0/0 = 1/B$).
**output** $(\pi^b, \underline{V}_1)$.

### F.2  Proof of Theorem A.2

The proof is completely analogous to that of Theorem 4 and Theorem A.1: we first establish the uniform convergence of the form (21), and then analyze the value functions similarly as in the proof of Theorem 4, except that we replace min over best response sets to max over best response sets, similar as in Theorem A.1. The guarantee we get has the same form as in Theorem 4 except that we replace $\phi$-functions by $\psi$-functions, and replace $\mathsf{gap}_\varepsilon$ with $\widetilde{\mathsf{gap}}_\varepsilon$. $\qquad \square$

## G  Proofs for Section B

### G.1  Proof of Theorem B.1

Recall that Algorithm 6 pulled each arm $(a, b) \in \mathcal{A} \times \mathcal{B}$ for $N = O(\log(AB/\delta)/\varepsilon^2)$ times, and $\widehat{\mu}_1(a, b), \widehat{\mu}_2(a, b)$ denote the empirical means of the observed rewards. By the Hoeffding inequality and union bound over $(a, b)$, with probability at least $1 - \delta$, we have

$$\max_{\pi^a \in \Delta_{\mathcal{A}}, b \in \mathcal{B}} |\widehat{\mu}_i(\pi^a, b) - \widehat{\mu}_i(\pi^a, b)| = \max_{(a,b) \in \mathcal{A} \times \mathcal{B}} |\widehat{\mu}_i(a, b) - \mu_i(a, b)| \leq \varepsilon/8 \quad \text{for } i = 1, 2. \tag{24}$$

**Properties of $\widehat{\mathsf{BR}}_{3\varepsilon/4}(\pi^a)$**  On the uniform convergence event (24), we have the following: for any $b \in \mathsf{BR}_{\varepsilon/2}(\pi^a)$, we have

$$\widehat{\mu}_2(\pi^a, b) \geq \mu_2(\pi^a, b) - \varepsilon/8 \geq \max_{b' \in \mathcal{B}} \mu_2(\pi^a, b') - 5\varepsilon/8 \geq \max_{b' \in \mathcal{B}} \widehat{\mu}_2(\pi^a, b') - 3\varepsilon/4,$$

and thus $b \in \widehat{\mathsf{BR}}_{3\varepsilon/4}(\pi^a)$. This shows that $\mathsf{BR}_{\varepsilon/2}(\pi^a) \subseteq \widehat{\mathsf{BR}}_{3\varepsilon/4}(\pi^a)$. Similarly we can show that $\widehat{\mathsf{BR}}_{3\varepsilon/4}(\pi^a) \subseteq \mathsf{BR}_\varepsilon(\pi^a)$. In other words,

$$\mathsf{BR}_\varepsilon(\pi^a) \supseteq \widehat{\mathsf{BR}}_{3\varepsilon/4}(\pi^a) \supseteq \mathsf{BR}_{\varepsilon/2}(\pi^a) \quad \text{for all } \pi^a \in \Delta_{\mathcal{A}}.$$

Notably, this implies that $\widehat{b} \in \widehat{\mathsf{BR}}_{3\varepsilon/4}(\widehat{\pi}^a) \in \mathsf{BR}_\varepsilon(\widehat{\pi}^a)$, the desired near-optimality guarantee for $\widehat{b}$.

**Near-optimality of $\widehat{\pi}^a$**  On the one hand, because $\widehat{\pi}^a$ approximately maximizes $\widehat{\mu}_1(\pi^a, \widehat{b}(\pi)^a)$ (in the sense of (13)), we have for any $\pi^a \in \Delta_{\mathcal{A}}$ that

$$\min_{b' \in \widehat{\mathsf{BR}}_{3\varepsilon/4}(\widehat{\pi}^a)} \widehat{\mu}_1(\widehat{\pi}^a, b') = \widehat{\phi}_{3\varepsilon/4}(\widehat{\pi}^a) \geq \widehat{\phi}_{3\varepsilon/4}(\pi^a) - \varepsilon/8 \overset{(i)}{\geq} \min_{b \in \mathsf{BR}_\varepsilon(\pi^a)} \widehat{\mu}_1(\pi^a, b) - \varepsilon/8,$$

**Algorithm 10** Subroutine `BestMixedLeaderStrategy`$(\widehat{\mu}_1, \widehat{\mu}_2)$

**Require:** Reward estimates $\widehat{\mu}_1, \widehat{\mu}_2 : \mathcal{A} \times \mathcal{B} \to [0, 1]$.
1: Define vectors

$$\widehat{v}_b = (\widehat{\mu}_1(a, b))_{a \in \mathcal{A}} \in [0, 1]^A, \quad \widehat{w}_b = (\widehat{\mu}_2(a, b))_{a \in \mathcal{A}} \in [0, 1]^A$$

for all $b \in \mathcal{B}$.
2: **for** $b \in \mathcal{B}$ **do**
3:     Solve the following linear program over $\pi^a \in \Delta_{\mathcal{A}}$:

$$\begin{aligned} \text{maximize } & (\pi^a)^\top \widehat{v}_b \\ \text{s.t. } & (\pi^a)^\top (\widehat{w}_b - \widehat{w}_{b'}) \geq 0 \quad \text{for all } b' \in \mathcal{B} \setminus \{b\}. \\ & \pi^a \in \Delta_{\mathcal{A}}. \end{aligned} \tag{25}$$

    Let $\widehat{\pi}^a(b), \widehat{u}(b)$ denote the solution and the value at the solution respectively.
4: Output $\widehat{b} = \arg\max_{b \in \mathcal{B}} \widehat{u}(b)$ and $\widehat{\pi}^a = \widehat{\pi}^a(\widehat{b})$.

---

where (i) is because $\widehat{\mathsf{BR}}_{3\varepsilon/4}(\pi^a) \subseteq \mathsf{BR}_\varepsilon(\pi^a)$. By the uniform convergence (24), we get that

$$\min_{b' \in \widehat{\mathsf{BR}}_{3\varepsilon/4}(\widehat{\pi}^a)} \mu_1(\widehat{\pi}^a, b') \geq \min_{b \in \mathsf{BR}_\varepsilon(\pi^a)} \mu_1(\pi^a, b) - \varepsilon/4 - 2 \cdot \varepsilon/8 \geq \phi_\varepsilon(\pi^a) - \varepsilon.$$

Since the above holds for all $\pi^a \in \Delta_{\mathcal{A}}$, taking the max on the right hand side gives

$$\sup_{\pi^a \in \Delta_{\mathcal{A}}} \phi_\varepsilon(\pi^a) - \varepsilon \leq \min_{b' \in \widehat{\mathsf{BR}}_{3\varepsilon/4}(\widehat{\pi}^a)} \mu_1(\widehat{\pi}^a, b') \overset{(i)}{\leq} \min_{b' \in \mathsf{BR}_{\varepsilon/2}(\widehat{\pi}^a)} \mu_1(\widehat{\pi}^a, b') = \phi_{\varepsilon/2}(\widehat{\pi}^a),$$

where (i) is because $\widehat{\mathsf{BR}}_{3\varepsilon/4}(\widehat{\pi}^a) \supseteq \mathsf{BR}_{\varepsilon/2}(\pi^a)$. In other words,

$$\phi_{\varepsilon/a}(\pi^a) \geq \sup_{\pi^a \in \Delta_{\mathcal{A}}} \phi_\varepsilon(\pi^a) - \varepsilon = \sup_{\pi^a \in \Delta_{\mathcal{A}}} \phi_0(\pi^a) - \mathsf{gap}_\varepsilon - \varepsilon.$$

This yields the first part of the bound for $\widehat{\pi}^a$.

On the other hand, since $\phi_\varepsilon(\pi^a)$ is increasing as we decrease $\varepsilon$, we directly have

$$\phi_{\varepsilon/2}(\widehat{\pi}^a) \leq \phi_0(\widehat{\pi}^a) \leq \sup_{\pi^a \in \Delta_{\mathcal{A}}} \phi_0(\pi^a).$$

This is the second part of the bound for $\widehat{\pi}^a$. $\qquad\qquad\qquad\qquad\qquad\qquad\qquad \square$

### G.2   Proof of Theorem B.2

We first check that the `BestMixedLeaderStrategy` subroutine is a correct algorithm for solving (14). Let

$$\widehat{V} = (\widehat{\mu}_1(a, b))_{a,b=1}^{A,B} \quad \text{and} \quad \widehat{W} = (\widehat{\mu}_2(a, b))_{a,b=1}^{A,B}$$

denote the matrix of estimated rewards. Observe that (14) is equivalent to the following problem

$$\begin{aligned} & \max_{\pi^a \in \Delta_{\mathcal{A}}} \max_{b \in \widehat{\mathsf{BR}}_{3\varepsilon/4}(\pi^a)} \widehat{\mu}_1(\pi^a, b) \\ = & \max_{b \in \mathcal{B}} \max_{\pi^a : \widehat{\mathsf{BR}}_{3\varepsilon/4}(\pi^a) \ni b} (\pi^a)^\top \widehat{V} e_b \\ = & \max_{b \in \mathcal{B}} \max_{\pi^a \in \Delta_{\mathcal{A}}} (\pi^a)^\top \widehat{V} e_b \\ & \text{s.t. } (\pi^a)^\top \widehat{W} e_b \geq (\pi^a)^\top \widehat{W} e_{b'} \quad \text{for all } b' \in \mathcal{B}, \end{aligned}$$

where $e_b \in \Delta_{\mathcal{B}}$ denotes the standard basis vector in $\mathcal{B}$ (1 at $b$ and 0 at $b' \neq b$). For each $b$, the above problem is exactly the same as the linear program (25). Then, the above problem requires maximizing the value over $b \in \mathcal{B}$, which is done in the output step of Algorithm 10. This shows that the `BestMixedLeaderStrategy` subroutine (Algorithm 10) is correct for solving (14). Note this

also proves that the $\arg\max$ in (14) is attainable (instead of the $\sup$ in Theorem B.1 which may not be attainable in general).

The rest of the proof is analogous to Theorem B.1 where we can again establish the uniform convergence (24) and obtain the suboptimality guarantee in terms of $\psi$ and $\widetilde{\mathsf{gap}}_\varepsilon$ instead of $\phi$ and $\mathsf{gap}_\varepsilon$, similar as in Theorem A.1. $\qquad\square$