# OpenReview forum: "Sample-Efficient Learning of Stackelberg Equilibria in General-Sum Games"
_NeurIPS.cc/2021/Conference — NeurIPS 2021 Poster_

### Official Review · Reviewer_79oc · 2021-07-08

**Rating:** 6
**Confidence:** 3

**Summary:**

The authors study the problem of learning Stackelberg equilibria from random samples in general sum games. The work will be of interest to both theoreticians of Stackelberg games and those who work on applications. The main contributions are:
- In bandit games, which are two-step Markov games where the leader and follower only observe their own rewards, the authors show that there is a constant gap between the optimal and the best strategy the leader can attain with a finite number of samples. This gap is a central feature of the paper—it exists in all settings that are analyzed by the authors, and the sample complexity results give you performance that approaches the best possible up to the gap. The authors give a tight bound on the number of samples required which is linear in the size of the action sets of the leader and follower and quadratic in the approximation ratio relative to the gap. The algorithm queries each leader action a fixed number of times (which is the same across all actions) to construct an estimate of the follower's best response function that has the desired properties.
- For bandit-RL games, where the leader selects an action that induces an MDP for the follower, the authors show that a quite similar algorithm can be applied. The main challenge is that the follower has to learn the optimal policy for each MDP (this is an RL setting with bandit feedback to the follower), and this requires an application of some recent work to yield appropriate estimates. The bandit-RL game setting reduces to bandit games when the horizon is of length 1. The derived sample complexity matches that of the bandit game for the horizon = 1 subcase, showing that the dependence on these parameters is tight. However, the scaling in horizon length is extreme—more than quintic. There is no known lower bound for the dependence on horizon.
- In linear bandit games, which are bandit games, but where the rewards are a linear function of the features of each action. The algorithm the authors state for this setting scales quadratically in the dimension of the feature space and quadratically in the approximation ratio. There is a factor of the dimension between this and the lower bound they derive.

**Limitations And Societal Impact:**

The authors don't spend a lot of time on limitations, but I found their treatment adequate—they identify gaps in knowledge (e.g., between upper and lower bounds) as well as fundamental limitations in the learning process.

Negative societal impact is not a concern for this theoretical paper.

**Main Review:**

I found the paper to be clear and well-written throughout, even in the supplemental material. The authors relate their work to prior work well and explain these connections extensively. The sample complexity results are original and useful to other researchers in this area. Bandit-RL games look like the most challenging of the settings and the most significant open questions come from this setting (the required dependence on states and horizon).

I don't have a firm basis for this, but I suspect the result about the constant gap for the bandit game setting was "known" in the Stackelberg general sum setting—it might be a consequence of other results or not stated directly. I suspect there is some kind of explanation in terms of the standard literature that could be added, but I don't have a citation. Even if this is the case, I don't think this is a major issue for the paper—I see the main contributions as analyzing the sample complexity "up to the gap."

The magnitude of the gap diminishes the interest of the authors' results to some extent—you would want some kind of guarantee that the gap would not be too large for a class of problems of interest.

Post-response edit:
- To clarify, I did not deduct any points for my suspicion that the constant gap result is "known" (more likely, a consequence of a known result). I do not have a specific reference for this and no reference was raised by the other reviewers either.
- The papers specific novelty are suspect. Specifically, 1. "To the best of our knowledge, this is the first result for sample-efficient learning of Stackelberg equilibrium in general-sum games with bandit feedbacks." (top of page 6) and 2. "This paper provides the first line of sample complexity results for learning Stackelberg equilibria in general-sum games." (conclusion, bottom of page 9). The pg. 9 claim is clearly false—the authors acknowledge other work in their related work section. The pg. 6 claim—I believe the authors intend to say that they are the first to provide results in their specific bandit feedback model, which is true to the best of my knowledge. Nevertheless, there are other bandit-style feedback models in the literature—for example, the learning model that assumes best responses by the adversary _is_ a bandit feedback model—it is just a different one from the authors'. These claims need to be revised.
- I still find the central claims of the paper to merit acceptance. The paper would benefit from integrating a stronger motivation about what kinds of games have a small gap—the point the authors made in their response.



**Time Spent Reviewing:**

4.5

---

> ### Author Response · Authors · 2021-08-10
> **Response to Reviewer 79oc**
>
> Thank you for the thoughtful feedback and comments!
>
> **“The magnitude of the gap diminishes the interest of the authors’ results to some extent---you would want the gap not be too large for a class of problems”**
>
> This gap in general exists in general-sum games in which the rewards $r_1$ (leader’s) and $r_2$ (follower’s) do not have any relations. However, we could expect the gap to be small if there are additional structural relationships between $r_1$ and $r_2$. As a simple example, in *zero-sum games* (where $r_1=-r_2$) with simultaneous plays and pessimistic tie-breaking, there is no such gap (${\rm gap}_\epsilon=0$). More generally, we could expect the gap to be small whenever there is some monotonicity structure between $r_1$ and $r_2$ (high $r_1$ implies high/low $r_2$ in some sense).
>
> On the other hand, when there is no such structure, this gap can be still large in general and poses an information-theoretic challenge to learning Stackelberg from noisy samples (as we show in Section 3). It reveals a unique distinction between noiseless versus noisy samples for learning Stackelberg equilibrium in general-sum games, which (to the best of our knowledge) has not been characterized in prior work.

---

> ### Author Response · Authors · 2021-09-05
> **Response to updated review**
>
> Thank you for the post-response edit and your positive feedback on our central contributions! We respond to your additional comments as follows.
>
> Re the 2nd point (concerns about our claims on Page 6 and Page 9): We acknowledge that our current claim on Page 9 is a bit imprecise. For both claims, we meant to claim what we wrote on Page 6, “first result for sample-efficient learning of Stackelberg equilibrium in general-sum games with bandit feedbacks”. Bandit feedback model in particular refers to those models where only the noisy reward of the actions played in each round are revealed. We believe alternative query models such as the best-response query are not considered or termed as a “bandit feedback model” (cf. Letchford et al. 2009 [26], Peng et al. 2019 [33]).
>
> We will revise the claim on Page 9 and clarify the terminology and ambiguities within these claims. We believe these do not affect our main contributions on learning Stackelberg equilibria with bandit feedback and sampling noise.
>
> Re the 3rd point (incorporate motivations on when the gap is small): Thank you for the suggestion; we will add more discussions about when the gap is small (along the lines of our earlier response) in the revision.

---

### Official Review · Reviewer_82ud · 2021-07-12

**Rating:** 6
**Confidence:** 3

**Summary:**

The paper considers learning Stackelberg equilibria in unknown games with noisy bandit feedback of the rewards obtained by leader and follower, respectively. The authors prove an information theoretic gap between the value of the true Stackelberg equilibrium and the one learned from finitely many data points. However, they show that approximate Stackelberg equilibria (defined according to a $\epsilon$-best response of the follower) can be learned efficiently using empirical reward estimates. Upper and matching lower bounds are provided on the number of required samples. Results are instantiated for bandit games, bandit reinforcement learning games, and linear bandit games.

**Limitations And Societal Impact:**

How do the obtained upper bounds compare with previous works, e.g. [26,33] which consider learning Stackelberg equilibria from best-response oracle? It is not clear which feedback model is less restrictive (since in practice we may not have access to the follower's reward and we could only observe the actions played), so I believe the obtained bounds deserve a fair comparison with these works.

Moreover, it would be nice to see some experimental results which can illustrate whether the derived bounds hold in practice.


**Main Review:**

The problem considered is relevant and I believe the contributions are novel, as previous work has mostly considered learning Nash equilibria. Moreover, the paper is well written and organized. Hence, my accept score.
However, I think that the authors should not claim (in Conclusions): "This paper provides the first line of sample complexity results for learning Stackelberg equilibria in general-sum game", since previous work has already considered learning Stackelberg equilibria under different assumptions (as discussed in related work section). Moreover, I believe some related works have been overlooked.
For instance, the StackelUCB algorithm by [Sessa et at., Learning to Play Sequential Games versus Unknown Opponents, NeurIPS 2020] also computes the optimal strategy for the leader, under noisy (albeit different) bandit feedback, by building an approximate best-response set.

**Time Spent Reviewing:**

3

---

> ### Author Response · Authors · 2021-08-10
> **Response to Reviewer 82ud**
>
> Thank you for your thoughtful feedback on our paper!
>
> **Related work of Sessa et al. (2020) “Learning to Play Sequential Games versus Unknown Opponents”**
>
> Thank you for pointing this out, we will properly cite this paper in our revision. There are two main differences between Sessa et al. (2020) and our work: (1) Sessa et al. (2020) assume that the opponent’s response function has a linear structure in a certain kernel space, whereas the best response function in our work does not necessarily possess any structure; (2) Sessa et al. (2020) consider additive observation noise in the action space, which is a different noisy feedback model from ours (noisy samples of the reward). These differences make the two settings incomparable and not solvable by each other’s algorithms. We will include these discussions in our revision.
>
>
> **“How to compare with previous works [26, 33] which considered learning Stackelberg from the best-response oracle… It is not clear which feedback model is less restrictive… the obtained bounds deserve a fair comparison with these works”**
>
> In general, the exact best-response query assumed in [26, 33] and the noisy bandit query in our paper are incomparable to each other. Our noisy bandit query is unable to produce exact best-responses for the follower no matter how many samples we observe, so our setting is not solvable by the algorithms in [26, 33]. On the other hand, exact best-responses also cannot yield full information about the individual reward values. Therefore, these two settings are in general incomparable and the bounds cannot be translated from one to the other. We will clarify this point in our revision.
>
> We also remark that our bandit feedback model is realistic for several real-world scenarios, such as the AI Economist, or more generally any large-scale game with a simulator. The feature about these games is that we can control both players, and can observe instantaneous rewards of any (joint) policy, but do not have direct access to---and thus have to learn---the follower’s best response function.

---

### Official Review · Reviewer_pAwo · 2021-07-16

**Rating:** 8
**Confidence:** 3

**Summary:**

The paper considers the problem of online learning the Stackelberg Equilibrium in general-sum games. The authors identify a fundamental gap between the exact and the estimated version of the Stackelberg value. Then they consider three general-sum games setting: bandit games, bandit-RL games, and linear bandit games, designing algorithms for these settings and providing sample complexity results.

**Limitations And Societal Impact:**

In my opinion, this paper has not particular limitations.

**Main Review:**

The paper studies an important problem in the online learning literature, analyzing the problem of learning Stackelberg Equilibrium in general-sum games. Although many works study this problem in the exact setting, the authors show that the problem becomes informationally-theoretically hard when we cannot observe the exact estimates of the rewards but only noisy samples. Moreover, the authors study the problem of learning epsilon-Stackelberg equilibrium in three settings: bandit games, bandit RL games, and linear bandit games. In my opinion, this paper shows important results for the game theory and online learning communities.
Moreover, these results leave out interesting future directions for the considered setting (for example, studying the optimal sample-complexity for bandit-RL games) and other settings as Leader-Follower MDPs, or Turn-based Markov Games.




**Time Spent Reviewing:**

3

---

> ### Author Response · Authors · 2021-08-10
> **Response to Reviewer pAwo**
>
> Thank you for your positive comments! We are glad you liked our paper and are very excited about this problem as well as these future directions.

---

### Official Review · Reviewer_c7dP · 2021-07-19

**Rating:** 6
**Confidence:** 4

**Summary:**

This paper provides sample complexity results and analysis of algorithms that learn Stackelberg equilibria for three types of games: general bandit games, bandit-RL games, and linear bandit games. The novelty of this paper lies in the fact that it studies a different solution concept, Stackelberg game, in general-sum games without assumptions regarding the convexity or concavity of the reward functions. To the best of my knowledge, this is the first paper that studies the sample complexity for learning Stackelberg equilibria under such games.

**Limitations And Societal Impact:**

Not applicable.

**Main Review:**

1. This paper focus on learning with bandit feedback for Stackelberg equilibria in static general-sum games, instead of general-sum stochastic (Markov) games. As the authors remarked, bandit games and bandit-RL games can be considered special cases of general-sum stochastic games (line 108). The significance of the paper is confined since only static games are considered (the bandit-RL is similar to a static game considering that the value functions are pre-computed using a reward-free learning algorithm.) It would be better if the authors can shed more light on (discuss related works about) learning Stackelberg equilibria for dynamic games (stochastic Stackelberg game and Stackelberg differential game) in which the leader and the follower make their decisions sequentially.


2. There exists literature that studies Stackelberg equilibria for stochastic games which the authors want to include in ‘related works’.
        *  Vasal, Deepanshu. "Stochastic Stackelberg games." arXiv preprint arXiv:2005.01997 (2020). .

3.The real-world application of the proposed algorithms is not well demonstrated. The authors did mention the optimal taxation problem by learning the Stackelberg equilibrium studied by another work [48], which is a problem that often times be formed as a Stackelberg game problem (Wang, You-Qiang. "Commodity taxes under fiscal competition: Stackelberg equilibrium and optimality." American Economic Review 89.4 (1999): 974-981.). This application is discussed at a problem formulation level. It is not clear how the results (sample complexity results) can offer insights for the potential applications. Hence, the current draft seems somewhat dry, overloaded with complexity metrics and math equations. Discussions in "Implications; Overview of Algorithms" does not offer two much practical interpretation. It requires effort for the reader on interpreting their practical meaning. It would be better if the authors can explain the results in the context of a practical application.

4. I did not thoroughly check all the proofs due to time constraint. I took a quick a look and no mistakes were identified.

**Time Spent Reviewing:**

5

---

> ### Author Response · Authors · 2021-08-10
> **Response to Reviewer c7dP**
>
> Thank you for your thoughtful feedback on our paper!
>
> **“This paper focuses on ‘static games’... could the authors shed more light on ‘dynamic games’ such as stochastic Stackelberg game and Stackelberg differential game”**
>
> If we understood correctly, by “dynamic games” you mean a general Markov game without special structures. We agree that this is an interesting and more general case. The focus of this paper is the new challenge of learning Stackelberg equilibrium of general-sum games from noisy bandit feedback, which we find already exhibiting some non-trivial phenomenon (existence of the “gap”) even in the “static games” such as bandit games and bandit-RL games. Therefore, we study these static games first in order to understand them thoroughly and best illustrate this phenomenon. That said, we agree that dynamics games would be an interesting and important direction for future work.
>
> **Related work of Vasal (2020) “Stochastic Stackelberg games”**
>
> Thank you for pointing out this related work; we will properly cite this in our revision. We remark some differences between the Vasal (2020) and our work: (1) Vasal (2020) considers the computational problem where full information of the game is observable, whereas we focus on the statistical aspect / sample complexities when we can only observe noisy samples of the game. (2) Vasal (2020) considers “conditionally independent controlled Markov processes”, which is another special case of Markov games, different from our bandit-RL games.
>
> **“Real-world application of proposed algorithms not well demonstrated… not clear how the results (sample complexity results) can offer insights for the potential applications… would be better if the authors can explain the results in the context of a practical application”**
>
> From a practical / applications perspective, our sample complexity results speak out how many samples are needed in order for our algorithm to learn an approximate Stackelberg equilibrium via noisy bandit feedback from the games. For example, in our bandit-RL game, our sample complexity $\tilde{O}(H^5SAB/\epsilon^2)$ scales polynomially with $H,S,A,B$ as well as $1/\epsilon$, where $H$ is the horizon of the game, $S$ is the number of states, and $A$, $B$ are the number of actions for the leader and follower, respectively. In the AI Economist example ([48] & our Section 2), $A$ is the number of actions (e.g. choices of tax rates) for the government, $H$ is the length of the game for the citizens, $S$ is the number of states within the game, and $B$ is the number of possible actions for the citizens. In this case, our results provide the first theoretical result that with a reasonably small amount of samples, our algorithm is guaranteed to learn the optimal policy (Stackelberg equilibrium) for the government.
>
> We will add more such practical interpretations in our revision.

---

### Decision · Program_Chairs · 2021-09-27

**Decision:**

Accept (Poster)

**Comment:**

The discussion mainly centered on the authors' claims of novelty.  While (to our knowledge) the technical contributions are indeed novel, there was consensus that the informal claims need to be significantly toned down -- there are too many variants of "initiates the theoretical study of sample-efficient learning of the Stackelberg equilibrium" and everything being "vastly open" which is simply not true given the references cited by the authors themselves.  (There is sometimes a qualifier about samples / a bandit model, but prior work could also be considered to involve samples and a bandit model -- this isn't exactly what sets this paper apart, though maybe the noise does, which is sometimes, but not always, mentioned in the qualifier.  As pointed out by one of the reviewers, there are also places where there isn't even any qualifier at all.)  The fact that this is claimed again and again, and the fact that this issue was apparently already raised in the ICML submission of an earlier version of this paper and apparently wasn't fully addressed (I wasn't involved in reviewing that version so I do not know the details), do honestly leave us a bit worried about just accepting the paper.  That seems unfortunate because it really wasn't necessary.  Still I think acceptance is the right decision, but the authors should appreciate that this really involves trusting them that they will truly fix this this time.  The same goes for any talk/poster about the material of course.

I wonder if the line of work on empirical game theory by Wellman and others should be cited here somewhere.